# Controlling target brain regions by optimal selection of input nodes

**Karan Kabbur Hanumanthappa Manjunatha**[1,2], **Giorgia Baron**[3], **Danilo Benozzo**[3], **Erica Silvestri**[3], **Maurizio Corbetta**[4,5,6], **Alessandro Chiuso**[3], **Alessandra Bertoldo**[3,6], **Samir Suweis**[1,6], **Michele Allegra**[1,6]*

**1** Physics and Astronomy Department "Galileo Galilei", University of Padova, Padova, Italy, **2** Modeling and Engineering Risk and Complexity, Scuola Superiore Meridionale, Napoli, Italy, **3** Information Engineering Department, University of Padova, Padova, Italy, **4** Neuroscience Department, University of Padova, Padova, Italy, **5** Venetian Institute of Molecular Medicine (VIMM), Padova, Italy, **6** Padova Neuroscience Center, University of Padova, Padova, Italy

* michele.allegra@unipd.it

**Data Availability Statement:** The EC matrices for individual subjects, as well as the code used to produce all figures in the manuscript, is made available at https://github.com/karankabbur/

## Abstract

The network control theory framework holds great potential to inform neurostimulation experiments aimed at inducing desired activity states in the brain. However, the current applicability of the framework is limited by inappropriate modeling of brain dynamics, and an overly ambitious focus on whole-brain activity control. In this work, we leverage recent progress in linear modeling of brain dynamics (effective connectivity) and we exploit the concept of target controllability to focus on the control of a single region or a small subnetwork of nodes. We discuss when control may be possible with a reasonably low energy cost and few stimulation loci, and give general predictions on where to stimulate depending on the subset of regions one wishes to control. Importantly, using the robustly asymmetric effective connectome instead of the symmetric structural connectome (as in previous research), we highlight the fundamentally different roles in- and out-hubs have in the control problem, and the relevance of inhibitory connections. The large degree of inter-individual variation in the effective connectome implies that the control problem is best formulated at the individual level, but we discuss to what extent group results may still prove useful.

## Author summary

Compared to healthy individuals, patients suffering from neurological diseases generally present widely altered brain activity patterns. A promising way to help these people restore a normal brain activity balance is using brain stimulation. As brain areas are interconnected in an intricate web, locally stimulating one or more brain areas can trigger effects across several distant locations, thus evoking a complex response. To achieve a specific response, one should know where (which stimulation sites) to stimulate. Several authors have proposed to solve this puzzle by using a computational model of brain activity together with control theory, a mathematical paradigm to design perturbations with desired effects on a dynamical system. Using an accurate model of brain activity fitted to experimental data from functional magnetic resonance imaging, we show that evoking

Effective-connectivity-and-Controllability-in-brain-dynamics.

**Funding:** This work was supported by the Department of Information Engineering of the University of Padova (Italy)(https://www.dei.unipd.it/home-page) DEI Proactive grant "Personalized whole brain models for neuroscience: inference and validation" to GB and DB; the Fondazione Cassa di Risparmio di Padova e Rovigo (CARIPARO)(https://www.fondazionecariparo.it/), Grant Agreement number 55403 to MC; the Ministry of Health, Italy (https://www.salute.gov.it/portale/home.html), "Brain connectivity measured with high-density electroencephalography: a novel neurodiagnostic tool for stroke- NEUROCONN" grant number RF-2008-12366899 to MC; the BIAL foundation (https://www.bial.com/), Grant Agreement number 361/18 to MC. The H2020 European School of Network Neuroscience (euSNN)(https://www.eusnn.eu/) H2020-SC5-2019–2, Grant Agreement number 869505 to MC; and the H2020 Visionary Nature Based Actions For Heath, Wellbeing & Resilience in Cities (VARCITIES) (https://varcities.eu/), H2020-SC5-2019–2, Grant Agreement 869505 to MC; the Ministry of Health, Italy (https://www.salute.gov.it/portale/home.html), "Eye-movement dynamics during free viewing as biomarker for assessment of visuospatial functions and for closed-loop rehabilitation in stroke (EYEMOVINSTROKE)", grant number RF-2019-12369300 to MC; the European Union (https://erc.europa.eu/homepage), "ERC-2022-SYG NEMESIS", Grant number 101071900 to MC. Views and opinions expressed are however those of the author(s) only and do not necessarily reflect those of the European Union or the European Research Council Executive Agency. Neither the European Union nor the granting authority can be held responsible for them. No funders funders played any role in the study design, data collection and analysis, decision to publish, or preparation of the manuscript.

**Competing interests:** The authors have declared that no competing interests exist.

arbitrary activity patterns in the whole brain requires stimulating a large number of brain areas simultaneously, which is unfeasible with current technology. One can nevertheless focus on a more affordable objective, controlling the activity of a small set of *target* regions. We discuss how to optimally select stimulation sites (so as to minimize the stimulation intensity) depending on the choice of the target regions, and on the structure of the brain connectivity network.

## Introduction

*Brain controllability* refers to the possibility of manipulating brain activity in a controlled way through external perturbations [1, 2], such as those that can be delivered non-invasively through neurostimulation techniques. For this goal, one can exploit control theory, a general mathematical framework to design perturbations of dynamical systems with a desired effect. In a first approximation, neural dynamics can be modeled as a linear and time-invariant [3], and one can try to control brain activity using the simple framework of linear network control theory [4]. The activity of the whole network can be controlled by acting on a subset of "driver nodes", and theory predicts which nodes should be selected and which input signal should be applied to obtain desired activity states. Since the first proposal by Gu et al. [5], this idea has been extensively explored [1, 2] and debated [6, 7].

So far, however, there has been limited success in directly applying this framework to predict the outcomes of neurostimulation experiments [8–10].]. In fact, the framework has been mainly applied in a relatively indirect way, by enriching the analysis of structural connectomes with a whole new set of tools based on *controllability metrics* [11]. The latter are node-wise metrics assessing the difficulty (energy cost) to reach desired states when specific nodes are selected as driver nodes, and they have proven very effective in summarizing features of the structural connectomes linked with cognitive function [12–20].

Among the obstacles hindering the practical applicability, and hence the widespread adoption of network control theory in neuromodulation experiments, a major one is a nearly exclusive focus on a quite ambitious objective, namely, whole-brain activity control. While a sufficiently well-connected network is in principle controllable with a single driver node, in practice a non-negligible fraction of the total number of nodes should be used as driver nodes to control the activity of the whole network with a realistic energy cost [21]. For large networks this means that many driver nodes are required. This is indeed the case for the brain: even with a coarse parcellation, $N \geq 60$ nodes are required to model the whole brain. However, current neurostimulation techniques such as TMS allow stimulating at most one (or two [22]) sites at the same time. Thus, a fine-grained control of whole brain activity is way beyond current experimental capabilities. A second relevant obstacle is the choice of the model for brain dynamics. The original proposal [5] assumed that the brain macroscopically follows linear dynamics with inter-areal couplings given by structural connectivity (SC), i.e., the amount of anatomical connections between areas estimated from diffusion MRI. This approach, however, was criticized by Tu et al. [6], who argued that couplings defined by structural connectivity miss important features of the dynamics. Dynamical coupling between brain areas is not simply proportional to anatomical connectivity: it can be asymmetric and include negative connections [23], whereas SC matrices inferred from diffusion imaging are always symmetric and positive. In fact, many authors have striven to develop powerful ways to fit functional MRI data at rest with a linear dynamical model and find the underlying *effective connectivity* (EC) structure [23–26].

In the present work, we propose a controllability approach relying on a realistic control objective and an EC-based dynamical model able to well fit the observed fMRI data. On one side, we will focus on a more affordable goal: *target control*, which consists in controlling only a selected group of regions [27] rather than the whole brain. On the other side, we will frame the control problem using EC matrices instead of SC matrices. EC at the individual level will be inferred from functional magnetic resonance imaging (fMRI) data through sparse dynamic causal modeling (sparse DCM) [26]. This model is a recent improvement over previous DCMs for resting state fMRI [3, 24], allowing for accurate parameter inference by combining lineariation of the hemodynamic response, discretization of the dynamics, and then a sparsity-inducing prior. Our proposal is illustrated by applying it to fMRI recordings of $N$ = 76 subjects from a large public database (Leipzig Study for Mind-Body-Emotion Interactions—LEMON dataset [28]). We will first confirm the main difficulties of whole-brain controllability already highlighted by previous literature [6, 7, 29], showing that the control cost (energy) scales exponentially with the number of target nodes. This will motivate our subsequent analysis of target control. We will initially discuss the simplest case of target control, where the goal is to control *a single target region* by acting on a remote brain region. Then, we will consider the case where one aims to control interconnected groups of regions defined by canonical resting state networks (RSNs). Controlling RSNs could be of clinical importance, as several brain diseases perturb specific RSNs [30, 31]. This analysis will also allow us to assess whether specific RSNs can be preferentially controlled from other RSNs, which could be of potential relevance in discussions of cognitive control. For both single target and RSN control, we will systematically address the problem of selecting good driver nodes (yielding a low energy cost) depending on the target, showing that centrality metrics can assist the choice of drivers, and discussing to what extent an individualized or a group selection is convenient.

The approach we propose has the potential to inform neurostimulation experiments with non-invasive techniques such as transcranial magnetic stimulation (TMS, [32]) or temporal interference [33]. These techniques could be used to control the activity of a (small) set of target regions, by inducing a desired activity pattern (i.e., activating or de-activating specific regions). To this aim, given estimates of each subject's effective connectivity obtained with fMRI, our approach allows identifying the optimal driver region (or set of driver regions) and assess the control energy (the amplitude of the control signal to be applied) necessary to control the target.

## Materials and methods

### Data collection

The resting-state fMRI dataset employed in this study consists of resting-state scans on a subset of 295 healthy subjects from the publicly available MPI-Leipzig Mind-Brain-Body dataset (LEMON) [28]. The data selection was performed on the original dataset (consisting of 318 individuals) by excluding participants with structural images heavily affected by artefacts or functional images with high head motion (less than 400 volumes with a mean framewise displacement < 0.4 mm) or affected by pre-processing failures and/or unavailability of rs-fMRI data [34]. While the first half of the dataset (147 subjects) was employed for clustering purposes (see details in the following sections), a final age- and gender-balanced sample of 76 individuals (younger: 20–39 M = 19, F = 19, older: 40–80 M = 19, F = 19) was extracted from the second half and then included in the controllability analysis of EC.

Data acquisition was performed with a 3T Siemens Magnetom Verio scanner, equipped with a 32-channel head coil. The protocol included a T1-weighted 3D magnetization-prepared 2 rapid acquisition gradient echoes (MP2RAGE; TR = 5,000 ms, TE = 2.92 ms, TI1 = 700 ms,

TI2 = 2,500 ms, first flip angle = 4˚, second flip angle = 5˚, FOV = 256 ×240 × 176 mm, voxel size = 1 × 1 × 1 mm, multiband acceleration factor [MBAccFactor] = 3), rs-fMRI scans (TR = 1,400 ms, TE = 39.4 ms, flip angle = 69˚, FOV = 202 × 202 mm, voxel size = 2.3 × 2.3 × 2.3 mm, volumes = 657, MBAccFactor = 4) and two spin echo acquisitions (TR = 2,200 ms, TE = 52 s, flip angle = 90˚, FOV = 202 × 202 mm, voxel size = 2.3 × 2.3 × 2.3 mm). During rs-fMRI scans, the subjects were asked to keep their eyes opened and to lie down as still as possible.

## Data preprocessing

For each control an individual pseudo-T1w image was obtained by multiplying the T1w 3D-MP2RAGE image with its second inversion time image and the structural preprocessing performed on this pseudo-T1w image included bias field correction (N4BiasFieldCorrection [35], skull-stripping (MASS [36]) and nonlinear diffeomorphic registration [37] to the symmetric MNI152 2009c atlas [38]. Pre-processing of rs-fMRI data consisted of slice timing (Smith et al. 2004), distortion (TOPUP [39]) and motion correction (MCFLIRT [40]) and nonlinear normalization to the symmetric MNI atlas [38] passing through the pseudo-T1w image via a boundary-based registration [41]. As a second step an ICA-based denoising was performed. The GIFT toolbox (http://trendscenter.org/software/gift/) was used to decompose the functional pre-processed data into independent components (ICs) by performing a group spatial-ICA as detailed in [42]. The ICs were classified into artifactual or resting-state network related in accordance with refs. [43, 44]. As a result, ICs that were related to banding artifacts, vascular or noise components were discarded. Then, 10 principal components related to CSF and white matter signal (5 from WM, 5 from CSF) were regressed out from rsfMRI timeseries as well as the 6 standard head motion parameters and their temporal derivatives. Then the denoised signal was high-passed with a filtering cut-off equal to 1/128 Hz.

## Parcellation and networks

We then extracted subject-level time series data from a 100-area parcellation scheme of the cortex provided by the Schaefer atlas [45], which maps to 7 resting-state functional networks: Control network (CON, 10 parcels), Default mode network (DMN, 16 parcels), Dorsal attention network (DAN, 9 parcels), Limbic network (LIM, 5 parcels), Saliency/Ventral attention network (VAN, 11 parcels), Somatomotor network (SMN, 6 parcels), Visual network (VIS, 5 parcels). We also defined a set of 12 subcortical and cerebellar regions based on the AAL3 segmentation [46]: for each hemisphere, 6 regions consisting of thalamus, caudate, putamen, pallidum, hippocampus and cerebellum (SUB, 12 parcels).

In addition, we assigned to each subject a binary temporal mask accounting for brain volumes corrupted by head motion ($FD > 0.4mm$) and we applied volume despiking to the time series by means of the icatb_despike_tc function of the GIFT toolbox. Moreover, the temporal traces were band-pass filtered (0.008 to 0.1 Hz).

Given the need to keep the computational load of sparse DCM at a reasonable level, a Consensus Clustering Evidence Accumulation (CCEA) procedure [47] was applied to reduce the number of cortical parcels derived from the Schaefer atlas. In order to account for hemodynamic differences across spatially distant parcels, this procedure was performed selectively for subsets of adjacent cortical regions referring to the same functional network. This additional constraint implied that only functionally homogeneous and spatially contiguous parcels could be grouped together, ensuring the consistency of hemodynamic properties of each cluster. The resulting clustering procedure provided 62 cortical clusters, from which demeaned fMRI time

courses (i.e., within-cluster mean BOLD signal) were extracted and supplied as inputs to sparse DCM together with the BOLD signals from subcortical sources.

## Sparse DCM

Dynamical Causal Modelling (DCM) was first introduced by *Friston et al.* [49]. It is a generative model of measured brain responses, where the output hemodynamic responses are evoked either by an underlying (unobserved) brain activity arising from experimental stimuli (during tasks) or spontaneous neural fluctuations (at rest). Here, we use the *sparse DCM* approach by Prando et al. [26]. This DCM variant implements a sparsity inducing mechanism that automatically prunes irrelevant connections, thereby avoiding the need to perform a selection between competing network structures. The algorithm has been further adjusted to account for the signal reliability of the temporal frames: high-motion frames (with framewise displacement $FD > 0.4$) were weighted less than normal frames ($FD \leq 0.4$) in the estimation algorithm. The model includes two layers: i) a coupled ODE system modeling neuronal activation $x(t)$, and ii) a mapping from neuronal activity $x(t)$ to the BOLD fMRI signal $y(t)$ (hemodynamic response). In formulas:

$$\dot{\mathbf{x}}(t) = A\mathbf{x}(t) + \boldsymbol{v}(t) \tag{1a}$$

$$\mathbf{y}(t) = h(\mathbf{x}(t); \theta_h) + \mathbf{e}(t) \tag{1b}$$

where $\mathbf{x}(t) = [x_1(t)\ldots x_n(t)]^T$ is the hidden neural activity of $n$ brain regions at time $t$, $A$ is the effective connectivity matrix (with matrix element $A_{ij}$ representing the effective connection from $j$ to $i$), $\boldsymbol{v}(t)$ is a stochastic term driving intrinsic brain fluctuations, $\mathbf{y}(t)$ is the BOLD fMRI response at time $t$, $\theta_h$ denotes collectively a set of biophysical parameters regulating the hemodynamic response (which is modelled with the Balloon-Windkessel model [49]), and $\mathbf{e}(t) \sim \mathcal{N}(0, R)$ is a Gaussian observation noise with covariance matrix $R$.

All model parameters, including the effective connectivity matrix $A$, need to be estimated by inverting the model on the measured fMRI data. To simplify the estimation procedure, Prando et al. [26] took two steps.

First, in a *discretization* step, justified by the low temporal resolution of fMRI scanners with sampling time $T_R \sim 0.5s$ to $3s$, the equation is integrated in steps of $T_R$. Measuring time in units of $T_R$, this leads to the finite difference equation

$$\mathbf{x}(k + 1) = e^A \mathbf{x}(k) + \mathbf{w}(k) \tag{2}$$

If we assume that the stochastic term $\boldsymbol{v}(t)$ is a white Gaussian noise with diagonal covariance matrix $\sigma^2 I_n$, then $\mathbf{w}(k)$ is also white Gaussian and its corresponding variance is given by $Q = \sigma^2 \int_0^1 e^{A\tau} e^{A^T \tau} d\tau$.

Second, in a *linearization* step the non-linear hemodynamic response is linearized by assuming a finite impulse response (FIR) for brain region $i$

$$y_i(k) = \sum_{l=0}^{s-1} h_{i,l} x_i(k - l) \tag{3}$$

where $h_i = [h_{i,0}, \ldots, h_{i,s-1}]^T$ are the FIR parameters for region $i$, with $s$ large enough to maintain temporal dependencies. The combination of these two simplifying moves reduces the model

to a linear stochastic model

$$\mathbf{x}^{(s)}(k+1) = A^{(s)}\mathbf{x}^{(s)}(k) + \mathbf{w}^{(s)}(k) \tag{4}$$

$$\mathbf{y}(k) = H^{(s)}\mathbf{x}^s(k) + \mathbf{e}(k) \tag{5}$$

where $\mathbf{x}^{(s)}(k) = [\mathbf{x}^T(k)\,\mathbf{x}^T(k-1)\ldots\mathbf{x}^T(k-s+1)]^T \in R^{n\times s}$ is the time-delayed activity signal, $A^{(s)}$ and $H^{(s)}$ are matrices containing the EC parameters ($A$)and the FIR parameters ($H$), respectively, $\mathbf{w}^{(s)}$ is a Gaussian noise terms with covariance matrix $Q^{(s)}$ (with blocks equal to $Q$), and $\mathbf{e}$ is a Gaussian noise with covariance matrix $R$.

The parameters $\theta = \{A, H, Q, R\}$, are estimated within a Bayesian framework by taking into account the observed values of the BOLD signal as well as the prior distribution of the parameters, chosen to be in this factorized form:

$$p(\boldsymbol{\theta}) \propto p_\gamma(A)p(Q)p(H)p(R) \tag{6}$$

Here, $p(Q)$ and $p(R)$ are uninformative priors, $p(H)$ is Gaussian (with means and variances fixed from knowledge of the typical hemodynamic responses [49]), and $p_\gamma(A)$ is a *sparsity inducing prior*,

$$p_\gamma(A) \sim \mathcal{N}(0, diag(\gamma_1, ..., \gamma_{n^2})) \tag{7}$$

Parameters are estimated by maximum a-posteriori estimates, using the expectation-maximization algorithm. The hyper-parameters $\gamma_i$ are estimated through marginal likelihood maximization, ensuring that a controlled fraction of the $\gamma_i$ are small and thus effectively inducing sparsity in $A$.

## Controllability

In our control framework, we neglect noise and assume that input is provided to a set of *driver nodes*. The system's dynamics become

$$\dot{\mathbf{x}}(t) = A\mathbf{x}(t) + B\mathbf{u}(t) \tag{8}$$

where $\mathbf{u}(t)$ is a time-dependent $r \times 1$ vector representing $r$ external inputs ($r \leq n$), $\mathbf{u}(t) = (u_1(t), ..., u_r(t))^T$ and $B$ is an $n \times r$ input matrix with which identifies the driver nodes, with $B_{ij} = 1$ if control input $u_j(t)$ is imposed on node $i$. The *Kalman's controllability condition* [50] states that the system is controllable if and only if the *controllability Gramian W*

$$W = \int_0^\infty dt e^{At}BB^T e^{A^T t} \tag{9}$$

$W$ is positive definite, $W > 0$ or $\lambda_{min}(W) > 0$ where $\lambda_{min}$ is the minimum eigenvalue. Due to numerical inaccuracies, it is impossible to assess whether an eigenvalue is exactly 0. Following common practice [51], we consider an eigenvalue to be 0 whenever is it below a very low numerical threshold $\epsilon = 10^{-12}$.

The *control energy* is defined as the (integrated) amplitude of the control signal used to steer the system from a given initial state $\mathbf{x}_0$ to a given final state $\mathbf{x}_f$ in a finite time $t_f$

$$E(u) = \int_0^{t_f} dt||\mathbf{u}(t)||^2 \tag{10}$$

Note that, if $t$ is measured in units of $T_R$, $E(u)$ is adimensional. The magnitude of $E(u)$ is related to the magnitude of the control signal, as $||\mathbf{u}|| \approx \sqrt{E/\tau}$ where $\tau$ is the time for which

$||u||$ is significantly different from 0. As all nonzero matrix elements of $B$ are of value 1, the magnitude of the term $B\mathbf{u}$ in Eq (8) is of order $\sqrt{E/\tau}$. This is to be compared with the magnitude of the initial and final states, $||\mathbf{x}_0||, ||\mathbf{x}_f|| = 1$. If $E(u) = 10^{12}$ and $\tau = 10^2$, this means that $||u|| \approx 10^5$, which means that the external driving must force the system through trajectories that pass through activity vectors of magnitude $10^5$ times larger than the initial and final activity vectors.

Let $\mathbf{u}^*$ be the optimal control input minimizing the control energy for a given pair $(\mathbf{x}_0, \mathbf{x}_f)$. In the limit $t_f \to \infty$, for normalized $(\mathbf{x}_0, \mathbf{x}_f)$, one has (see S1 Text for details)

$$E(\mathbf{u}^*) \leq \mathcal{E} \equiv \frac{1}{\lambda_{min}(W)} \tag{11}$$

where the $\lambda$'s are simply the eigenvalues of $W$. A common metric to assess the difficulty of steering the system is the upper bound $\mathcal{E}$, which gives the control energy required to steer the system along the worst possible eigendirection of the Gramian $W$.

In *target control* [27], one aims to control only a selected subset of target nodes. Let $\mathcal{T} = \{c_1, c_2, c_3...c_S\}$ be the target node set (of cardinality $S$) and let

$$\mathbf{y}(t) = C\mathbf{x}(t) \tag{12}$$

be the output vector describing the activity of the the target nodes we want to control $(\mathbf{y}(t) \in \mathbb{R}^S)$, with $C_{ij} = 1$ if and only if $i = j$ and $j \in \mathcal{C}$. The definition of target controllability follows from that of standard (Kalman) controllability, where the system is now defined by the triple $(A, B, C)$ instead of the pair $(A, B)$ [27]. The system $(A, B, C)$ is *target controllable* with respect to target node set $\mathcal{C}$ if the target controllability Gramian

$$W_C = CWC^T \tag{13}$$

is positive definite. Similar to the one we have seen in the case of full controllability, we have for the control energy (see S1 Text)

$$E(\mathbf{u}^*) \leq \mathcal{E}^{target} = \frac{1}{\lambda_{min}(W_C)} \tag{14}$$

If a single driver node $i$ is used, and the target is in turn a single node $j$, the expression of the control energy significantly simplifies. We have $B = \mathbf{e}_i$ and $C = \mathbf{e}_j^T$, where $\mathbf{e}_i$ is the $i$-th canonical basis vector. Thus

$$\mathcal{E}_{i \to j} \equiv E_{min}^{target} = \left(W_{jj}^{(i)}\right)^{-1} = \frac{1}{\int_0^\infty dt \left[e^{At}\right]_{ij}^2} \tag{15}$$

To better highlight the controllability properties of each node within a network, we define two quantities, the *driver centrality* and the *target centrality* by averaging the pairwise control energy over all possible targets and all drivers, respectively:

$$\mathcal{E}_i^d = \frac{1}{n} \sum_j \mathcal{E}_{i \to j} \tag{16a}$$

$$\mathcal{E}_i^t = \frac{1}{n} \sum_j \mathcal{E}_{j \to i} \tag{16b}$$

## Centrality measures and shortest paths

**EC centralities.** A possible way to select driver nodes is based on centrality measures computed on the graph defined by the EC matrix $A_{ij}$. Nodes of the network are ranked according to a selected centrality measure, and high-ranking nodes are selected as driver nodes. A viable set of centrality measures appropriate for this approach is the following, which includes both on classical and controllability-tailored measures i) Out-strength (sum of absolute strength of outgoing connections) $\mathcal{A}_i^{out} = \sum_j |A_{ji}|$ and in-strength (sum of absolute strength of incoming connections) $\mathcal{A}_i^{in} = \sum_j |A_{ij}|$; note that $\mathcal{A}_i^{out} \neq \mathcal{A}_i^{in}$ in general because EC matrices are non-symmetric, and we take absolute values since entries of $A$ can have both positive and negative sign ii) Page rank [52], which determines the nodes centrality based on how long a random walk can remain "stuck" in the node; in this case, nodes are ranked in decreasing order (to avoid issues related to the presence of negative weights in the computation of PageRank, we considered an unweighted version of the network replacing all nonzero links with ones) iii) The ratio of absolute out-strength and in-strength $r_w(i) = \sum_{j=1}^{N} |A_{ji}| / \sum_{j=1}^{N} |A_{ij}|$, which was argued to be a good centrality measure to select driver nodes in the context of controllability [53], iv) the control centrality proposed by Lindmark et al [54], $r_i = p_i/q_i$. In the last centrality measure, $p_i = Tr(W^{(i)})$, where $W^{(i)} = \int_0^\infty e^{At}\mathbf{b}_i\mathbf{b}_i^T e^{A^T t}$ is the controllability Gramian corresponding to using node $i$ as a driver, with $\mathbf{b}_i$ the $i$-th column of $B$; $q_i = Tr(M^{(i)})$, where $M^{(i)}$ is the observability Gramian $M^{(i)} = \int_0^\infty e^{A^T t}\mathbf{e}_i\mathbf{e}_i^T e^{At}$. Respectively, $p_i$ and $q_i$ measure the ability to control other nodes from node $i$, and the ability to control node $i$ indirectly from other nodes.

**Energy centralities.** Based on the single-driver-single-target energy (15) we define two quantities, the *driver centrality* $\mathcal{E}_i^d = \sum_j \mathcal{E}_{i \to j}$ and the *target centrality* $\mathcal{E}_i^t = \sum_j \mathcal{E}_{j \to i}$, by summing the pairwise control energy over all possible targets and all drivers, respectively. The driver energy represents the average energy with which we can control another node, using node $i$ as a driver. The target energy represents the average energy with which we can control node $i$ using another node as a driver.

**Shortest paths.** In the graph defined by $A_{ij}$, we defined the length of a path by summing the length of each edge, assigning to the edge between nodes $k$ and $l$ a length $1/|A_{kl}|$, i.e., inversely proportional to the effective connection between $k$ and $l$. We can thus compute shortest paths in the graph through Dijkstra's algorithm [55]. We denote by $\ell_{ij}$ the length of the shortest path between nodes $i$ and $j$.

## Optimal node placement and rank aggregation

For a given subset of target nodes $\mathcal{T}$, we rank nodes according to different centralities, and select as driver nodes the $n_d$ nodes with highest rank. Then, we can identify which centrality allows achieving the lowest value of control energy ('optimal centrality'), rank nodes according to the optimal centrality, and select as drivers the $n_d$ driver nodes with highest rank ('optimal drivers'). In a fist step, we can identify optimal drivers for each subject independently, yielding a subject-dependent set of optimal drivers $\mathcal{O}(s)$ where $s$ is a subject index.

Then, given a certain set of nodes $\mathcal{D}$, non-overlapping with $\mathcal{T}$ ($\mathcal{D} \cap \mathcal{T} = \emptyset$), we can test whether optimal drivers are preferentially selected from $\mathcal{D}$. This problem is analogous to the problem where one has balls of two colors (blue/red) divided in two urns (A/B), and wants to test whether urn A contains an anomalous proportion of blue balls (i.e., statistically unlikely if balls are randomly placed in the two urns regardless of color). This problem can be solved performing a Fisher exact test [56]. Here, we have optimal/non-optimal nodes (belonging respectively to $\mathcal{O}$ and its complement $\bar{\mathcal{O}}$), divided in two sets. To know whether optimal nodes are

preferentially selected from $\mathcal{D}$, we perform a Fisher exact test on the quantities

$$n_1 = \sum_s |\mathcal{O}(s) \cap \mathcal{D}|, \quad n_2 = \sum_s |\mathcal{O}(s) \cap \bar{\mathcal{D}}|, \quad n_3 = \sum_s |\bar{\mathcal{O}}(s) \cap \mathcal{D}|, \quad n_4 = \sum_s |\bar{\mathcal{O}}(s) \cap \bar{\mathcal{D}}|,$$

which correspond to the number of optimal/non-optimal drivers that belong/not belong to $\mathcal{D}$.

One can use rank aggregation to obtain a group-wise set of optimal nodes. Among the possible approaches to rank aggregation [57], we used the most basic approach, namely, computing the average rank (other common criteria such as Borda and Dowdall [57] give very similar results). For each subject, we thus rank nodes according to the optimal centrality, produce a group ranking using rank aggregation, and select as drivers the $n_d$ driver nodes with highest rank.

## Results

### Effective connectivity matrices

We considered resting state fMRI data of $N = 76$ participants, parcellated into $n = 74$ regions (58 cortical regions + 16 subcortical regions). Applying sparse DCM [26] to the regional time series, we obtained individual effective connectivity matrices $A$. The matrix element $A_{ij}$ represents the directed connection (influence) from $j$ to $i$. The linear model given by DCM obtained a very good fit of the data, with a correlation between the functional connectivity (FC, standard Pearson correlation matrix between the BOLD signals of all areas) of the model and the actual FC of 0.78 (on average over subjects). Example matrices are shown in S1 Fig. The EC matrices have nonzero, negative diagonal entries, $A_{ii} < 0$, as required for dynamic stability [49]. Effective connections are sparse: on average over all subjects, the link density was 0.39 (61% of matrix entries are zero). On average 59.9% of links were positive ("excitatory"), and 40.1% negative ("inhibitory"). The EC matrices exhibit a large inter-subject variability. To assess the degree of inter-subject consistency, we evaluated the Pearson correlation between the EC matrices of all pairs of subjects: on average over all pairs, the correlation was 0.49 (s.d. 0.03). This figure is comparable with the inter-subject consistency of FC matrices (average 0.49, s.d. 0.08).

### Scaling of control energy with the number of driver and target nodes

Two key parameters determining the control energy (Eq 11) are the number of driver nodes, $n_d$, and the number of target nodes, $n_t$.

**Scaling with the number of driver nodes.**   In the literature, the case $n_t = n$ is usually considered, where one tries to control the whole network. We thus first fix $n_t = n$ and analyze the control energy $\mathcal{E}$ as a function of the number of driver nodes $n_d$ (Fig 1A).

For each value of $n_d$, we selected driver nodes as high-ranking nodes according to different centrality measures (Methods). Results for a random selection of nodes are also presented. The control energy is exceedingly high ($> 10^{12}$) unless a significant ($\gtrsim 15\%$) fraction of the nodes are used as driver nodes. This result is fully in agreement with the previous results of Tu et al. [6] and resonates with previous theoretical controllability studies. On one side, since the matrix $A$ has nonzero diagonal entries, the maximum matching theorem ensures that the system is controllable by applying a single external input jointly to all nodes, i.e. $B = [1, 1, 1, \ldots, 1]^T$ [21, 58]. However, when computing the minimum eigenvalue of the corresponding Gramian matrix, we systematically obtain very small values (of the order $10^{-13}$). Therefore, this simple control solution is not applicable in practice. In fact, as highlighted in ref. [21], unless a considerable fraction of the nodes are controlled, control energy is astronomically large, and control trajectories are extremely long and numerically unstable. Fig 1B shows the distribution

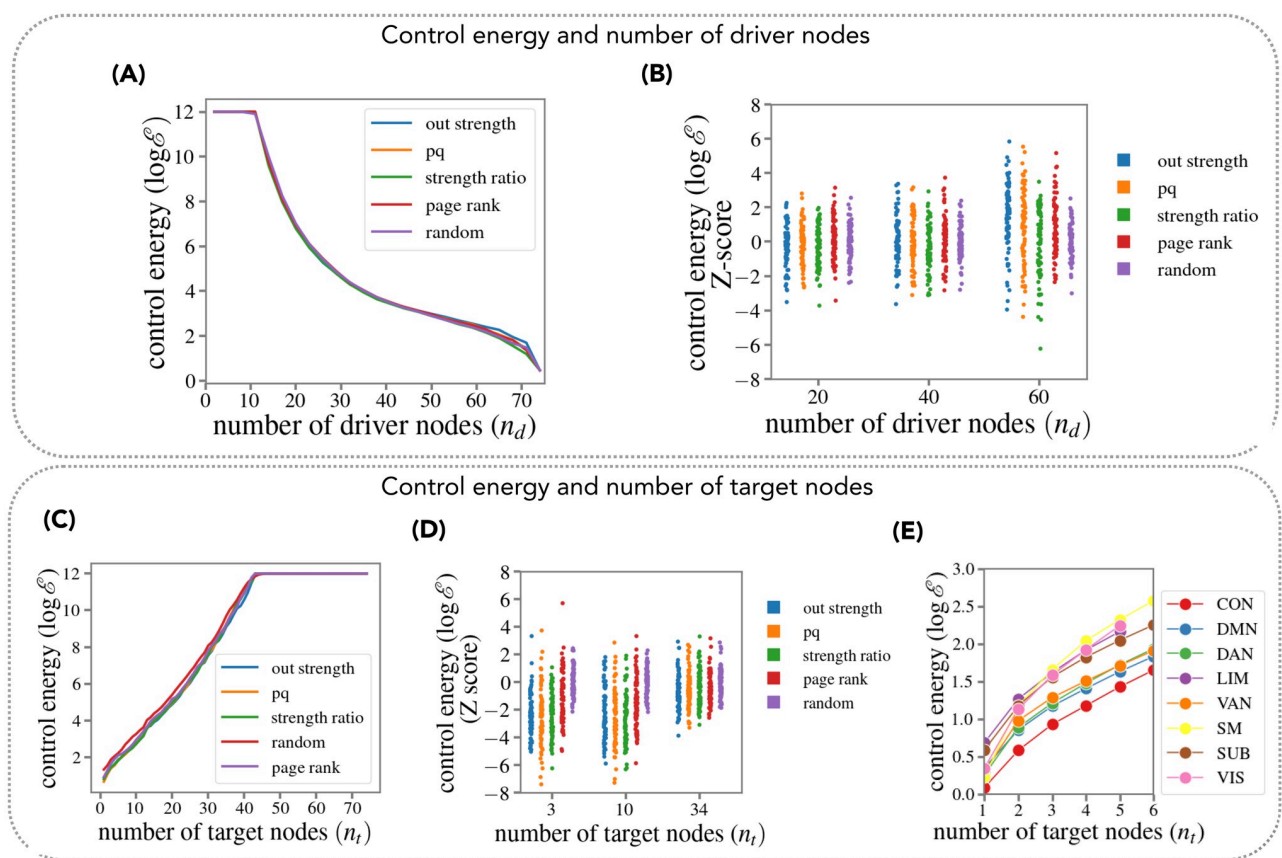

**Fig 1. Dependence of the control energy on the number of driver and target nodes. (A)** Energy to control the whole brain network (median over subjects) as a function of the number of driver nodes $n_d$. For each $n_d$, nodes were selected based on a ranking of centrality measures. **(B)** Energy to control the whole brain network (distribution over subjects), for three values of $n_d$. For each subject, energy values were z-scored with respect to the mean of the random node selection. **(C)** Energy to control a varying number of target nodes, using $n_d = 5$ driver nodes selected according to different centrality measures as well as randomly. Lines represent the average control energy over subjects (over both subjects and realizations for the random curve) **(D)** Energy to control target nodes, using $n_d = 10$ driver nodes (distribution over subjects). For each number of target nodes, energy values were z-scored with respect to the mean energy obtained with the random node selection. **(E)** Energy to control a varying number of target nodes within each of 8 RSNs, using $n_d = 10$ driver nodes selected according to a ranking based on the out-strength. In (A-D), all random curves were obtained by averaging over $M = 100$ random selections of driver nodes.

(over subjects) of control energies obtained with different (centrality-based) choices of driver nodes, with energies z-scored to the mean of the distribution obtained with a random choice of driver nodes. The control energy depends quite weakly on the choice of driver nodes, with centrality measures not affording any significant advantage over a random choice of nodes.

**Scaling with the number of target nodes.** Given the difficulties with whole-brain network controllability, we next consider the dependence of the control energy on the total number of target nodes $n_t$. In Fig 1C we plot the energy required to control a varying number of target nodes. Here, target nodes were chosen randomly (nodes were randomly sorted and an increasing number of nodes was included in the set of target nodes following the sorting). We used $n_d = 5$ driver nodes selected according to 4 different centrality measures, as well as randomly. The control energy scales *exponentially* with $n_t$. Since current techniques allow perturbing only one or a few nodes simultaneously, this implies that the control problem is practically solvable only for a low number of target nodes. The strictly exponential scaling depends on the fact that target nodes were chosen randomly: therefore, target nodes were on

average not strongly connected to driver nodes. If one restricts attention to groups of strongly connected nodes, such as those belonging to the same resting state network (RSN), we observe a deviation from the exponential scaling (Fig 1E). In particular, the scaling of $\log \mathcal{E}$ with $n_d$ is weakly sublinear, showing a weak "saturation effect" whereby adding new nodes to the set of target nodes is progressively less costly. Fig 1C also shows that selection of driver nodes has an effect on the control energy. In particular, the random choice systematically yields larger energies than centrality-based choices. In Fig 1D we show the distribution (over subjects) of the (log-)control energies, z-scored to the mean of the distribution obtained with a random choice of driver nodes. For all values of $n_t$, a centrality-based choice of driver nodes affords a significant advantage over a random choice of driver nodes (T-test, $T(75) < -8$, $p < 10^{-10}$ corrected for 5 multiple comparisons). The effect is more pronounced for low $n_t$. These results do not depend either on the specific $n_d$ used (analogous results are obtained with $n_d = 5$, $n_d = 20$). We stress that centrality measures are computed directly on the EC, not on standard FC. In S9(A) Fig we show the overlap between the 10 highest-ranking nodes selected with the EC and with the FC, respectively. On average (over subjects), the overlap was only 15% (for out degree), 11% (for pq-centality), 13% (for degree ratio centrality), 28% (for page rank centrality). Correspondingly, in S9(B) Fig we show the distribution of the (log-)control energies obtained using FC-based instead of EC-based centrality measures. The values of the (log-)control energy were z-scored to the mean of the distribution obtained with an EC-based choice of driver nodes. The FC-based choice consistently yields larger energies than the EC-based one.

*In summary, the control energy scales exponentially with the number of target nodes. When the number of target nodes is large, the energy is exceedingly large unless a significant fraction of the nodes is used as driver nodes. These results imply that whole-brain controllability is unfeasible with current techniques. The dependence of the control energy on the choice of driver nodes is appreciable for a low number of driver and target nodes.*

## Single-node target controllability

Given the unfeasibility of whole-brain controllability, in the remainder we concentrate on target controllabilty of selected brain regions or groups of regions. We first consider the simplest, and experimentally most viable target controllability problem: controlling *a single target node* by using *a single driver node*. This case corresponds to the typical experimental setting where one wishes to activate/deactivate a specific brain region by stimulating a (single) remote region. Furthermore, it allows clarifying general relations between effective connectivity matrices and controllability.

**Which connections contribute to control.** The control energy for the single-driver-single-target case is given by Eq (15), which determines the energy $\mathcal{E}_{i \to j}$ required to control node $j$ through node $i$. In Fig 2A we plot $\mathcal{E}_{i \to j}$ against $A_{ji}$ for a single subject. Unsurprisingly, for positive links ($A_{ji} > 0$, blue) $\mathcal{E}_{i \to j}$ is negatively correlated with $A_{ji}$ (Spearman $R \approx -0.39$ for this subject).

This means that if there is a large effective connection between $i$ and $j$, it is less costly to control $j$ through $i$ (control energy decreases). However, for negative connections ($A_{ji} < 0$) $\mathcal{E}_{i \to j}$ is *positively* correlated with $A_{ji}$ (Spearman $R \approx 0.44$ for this subject). Thus, negative connections have a positive, not a detrimental effect for controllability: if there is a large, negative effective connection between $i$ and $j$, it is less costly to control $j$ through $i$. Stated otherwise, effective connections reduce the required control energy with a contribution dependent on their strength, but independent of their sign. Group results confirm this finding (Fig 2E). The average Spearman correlation between EC and control energy is $R = -0.40 \pm 0.03$ (mean ± s.d.) for positive connections and $R = 0.41 \pm 0.05$ for negative connections. From Fig 2A we also see

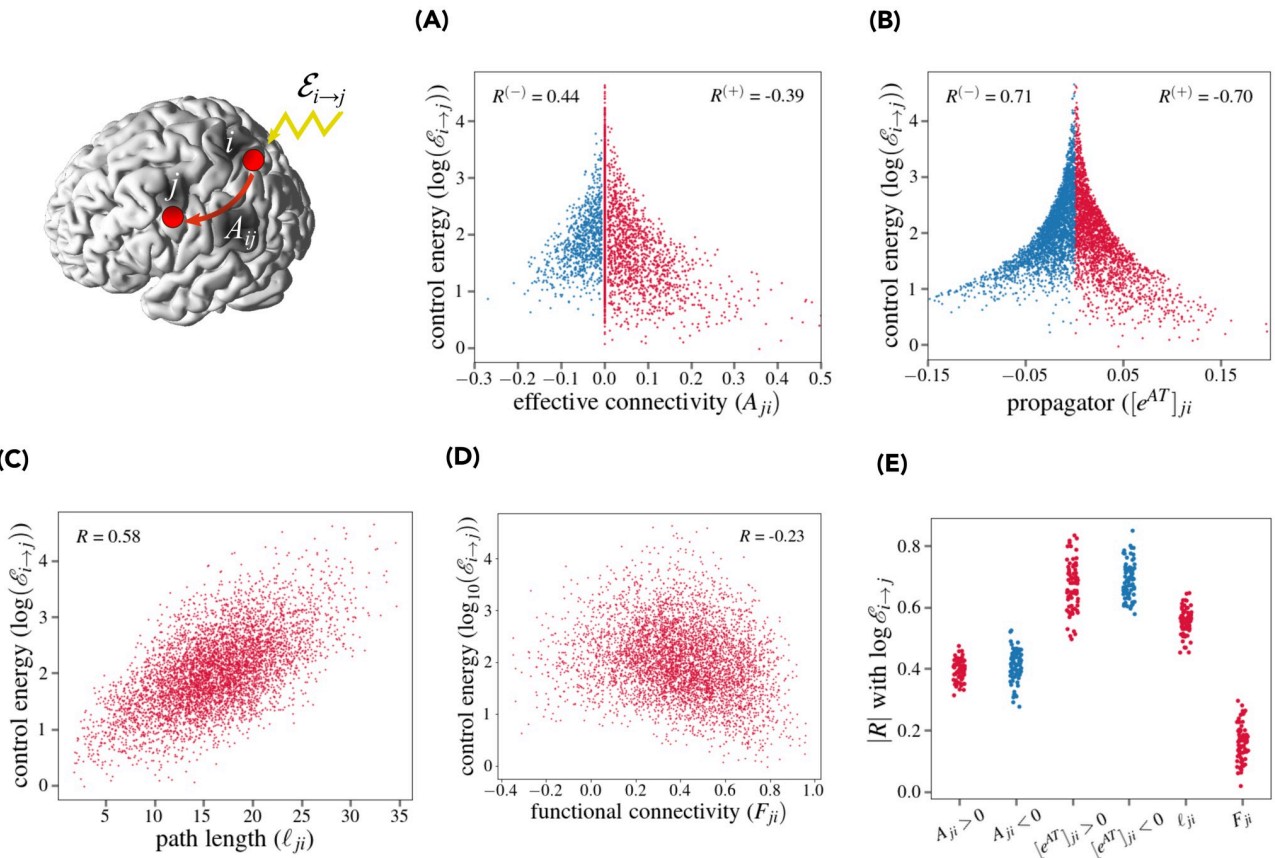

**Fig 2. Relation between single-driver single-target energy and effective connectivity.** We considered the control energy required to consider a single target $j$ using a single target $i$ (top left; the brain image was visualized with BrainNetViewer [48]). **(A)** Control energy $\mathcal{E}_{i \to j}$ (energy required to control a single node $j$ using a single node $i$ as driver) vs. the effective connectivity between $i$ and $j$, $A_{ji}$ for a single representative subject. Positive ($A_{ji} > 0$) and negative ($A_{ji} < 0$) effective connections are highlighted in blue and red respectively. The value of Spearman correlation between $A_{ji}$ and $\log \mathcal{E}_{i \to j}$ is shown for positive ($R^{(+)}$) and negative ($R^{(-)}$) connections respectively. **(B)** Control energy $\mathcal{E}_{i \to j}$ vs. the absolute value of the $i,j$ matrix element of the propagator $e^{AT}$ for $T = 10$ for a single representative subject. Positive ($[e^{AT}]_{ji} > 0$) and negative ($e_{ji}^{AT} < 0$) effective connections are highlighted in blue and red respectively, along with the corresponding values of Spearman correlation with $\log \mathcal{E}_{i \to j}$. **(C)** Control energy $\mathcal{E}_{i \to j}$ vs. the length of the shortest path $\ell_{ji}$ connecting $i$ and $j$ using effective connections for a single representative subject. **(D)** Control energy $\mathcal{E}_{i \to j}$ vs. the $i,j$ matrix element of the functional connectivity $F$ for a single representative subject. **(E)** Distribution (over subjects) of the absolute value of Spearman correlation $|R|$ between $\mathcal{E}_{i \to j}$ and $A_{ji}$ (positive and negative connections), $[e^{AT}]_{ji}$ (positive and negative connections), $\ell_{j,i}$, and $F_{ji}$.

that large effective connections are a sufficient, yet not necessary condition to have low control energy. We hypothesized that this is due to indirect connections. Indeed, mathematically, the influence of node $i$ onto $j$ over a time scale $t$ is exerted though the propagator $e^{At}$, rather than $A$. The matrix element $[e^{At}]_{ji}$ effectively integrates the effect of direct and indirect paths between $i$ and $j$. In Fig 2B we plot $\mathcal{E}_{i \to j}$ against $[e^{At}]_{ji}$ for a single subject, where $T = 10$ (corresponding to 14 seconds, as time is measured in units of $TR = 1.4s$). This is the timescale required for a local perturbation to fully propagate to distal nodes according to the DCM model (S2 Fig). We obtain stronger correlations ($R \approx -0.70, R \approx 0.71$ for positive and negative connections respectively). Over all subjects (Fig 2E), we obtain $R = 0.67 \pm 0.08$ for positive connections and $R = 0.69 \pm 0.06$ for negative connections. To strengthen the conjecture that the value of the control energy $\mathcal{E}_{i \to j}$ is related to the presence of direct and indirect connections between $i$ and $j$, in Fig 2C we plot $\mathcal{E}_{i \to j}$ against $\ell_{ji}$, the length of the shortest path between $i$ and $j$

in the graph defined by $A_{ji}$. We observe a strong positive correlation ($R \approx 0.58$): if nodes $i$ and $j$ are "near" (i.e., linked by strong direct or indirect connections), the control energy is lower. Over all subjects (Fig 2E), the average correlation coefficient is $0.56 \pm 0.04$. Finally, we note that the value of the control energy $\mathcal{E}_{i \to j}$ is poorly predicted by the standard functional connectivity $F_{ji}$ between nodes $i$ and $j$ (Fig 2D), as we observe only a weak negative correlation ($R \approx -0.23$). Over all subjects (Fig 2E), the average correlation coefficient is $-0.16 \pm 0.06$. *In summary, the presence of large (direct and indirect, positive or negative) connections between i and j determines a low control energy $\mathcal{E}_{i \to j}$.*

**Optimal driver and target nodes.** Based on the results of the previous section, we assumed that nodes with strong incoming connections would require a low energy to be controlled, and nodes with strong outgoing connections would require low energy to control other nodes. We verified this hypothesis by computing the link between the driver/target centrality $\mathcal{E}_i^d, \mathcal{E}_i^t$ of a node (Methods), representing the average energy when using a node as a driver or target, and the in- and out-strength of that node $\mathcal{A}_i^{in}, \mathcal{A}_i^{out}$. On average over subjects (S3(A) Fig), $\mathcal{E}_i^d$ is strongly negatively correlated with $\mathcal{A}_i^{out}$ ($R = -0.71 \pm 0.05$, mean $\pm$ s.d.) but uncorrelated with $\mathcal{A}_i^{in}$ ($R = 0.00 \pm 0.15$). Conversely, $\mathcal{E}_i^t$ is strongly negatively correlated with $\mathcal{A}_i^{in}$ ($R = -0.72 \pm 0.07$) but weakly correlated with $\mathcal{A}_i^{out}$ ($R = -0.10 \pm 0.17$). This implies that asymmetries between incoming and outgoing connections have large significance for control. These asymmetries can be appreciated when considering EC (which is non-symmetric), but not standard functional connectivity, FC (which is by definition symmetric). In fact, when considering the FC strength ($\mathcal{F}_i = \sum_j F_{ji}$ where $F$ is the functional connectivity matrix), we did not find any relation with either the driver centrality ($R = -0.02 \pm 0.12$), or the target centrality ($R = 0.17 \pm 0.13$). In S4(A) Fig we illustrate the conceptual difference between in-hubs (nodes with strong incoming connections) and out-hubs (nodes with strong outgoing connections), while in S4(B)–S4(D) Fig we report brain plots showing the in-strength and out-strength of different regions, as well as the total FC strength. Our results suggest that "in-hubs" of the effective connectivity are the easiest nodes to control, while "out-hubs" of effective connectivity are the best nodes to use to control other nodes, and should possibly be chosen as driver nodes. Intuitively, as the control signal must reach the target node from the driver node, the easiness to control a given target node is related to the existence and the number of network paths leading from the driver to the target. *In summary*, on average in-hubs are more easily reached (there are many paths leading to them), out-hubs are the best drivers (many paths from them lead to the possible targets). Note that in- and out-hubs of EC do not trivially align with FC hubs.

An important *caveat* is that out-hubs are not strongly consistent across subjects. In fact, we assessed the consistency of in- and out-strength over subject by computing the coefficient of variation $c_V$ (standard deviation/mean). Small values imply high consistency and vice versa. We obtained: for out-strength $c_V = 0.31 \pm 0.07$ (mean $\pm$ st. dev. over nodes), for in-strength $c_V = 0.17 \pm 0.03$ (S3(B) Fig). Thus, the in-strength is much more consistent than the out-strength (T test, $T(73) = -19.2$, $p < 10^{-30}$). Correspondingly, the target centrality ($c_V = 0.10 \pm 0.02$) is much more consistent than the driver centrality ($c_V = 0.19 \pm 0.03$; T test, $T(73) = -19.5$, $p < 10^{-30}$). Thus, the easy-to-control nodes are more consistent across subjects than the good "input nodes" by which one can control other nodes.

To identify good driver nodes, we ranked nodes based on $\mathcal{E}_i^d$. Fig 3A shows a rendering of the brain, with region color corresponding to the average rank (average over subjects) based on $\mathcal{E}_i^d$.

**(A)**                    **(B)**

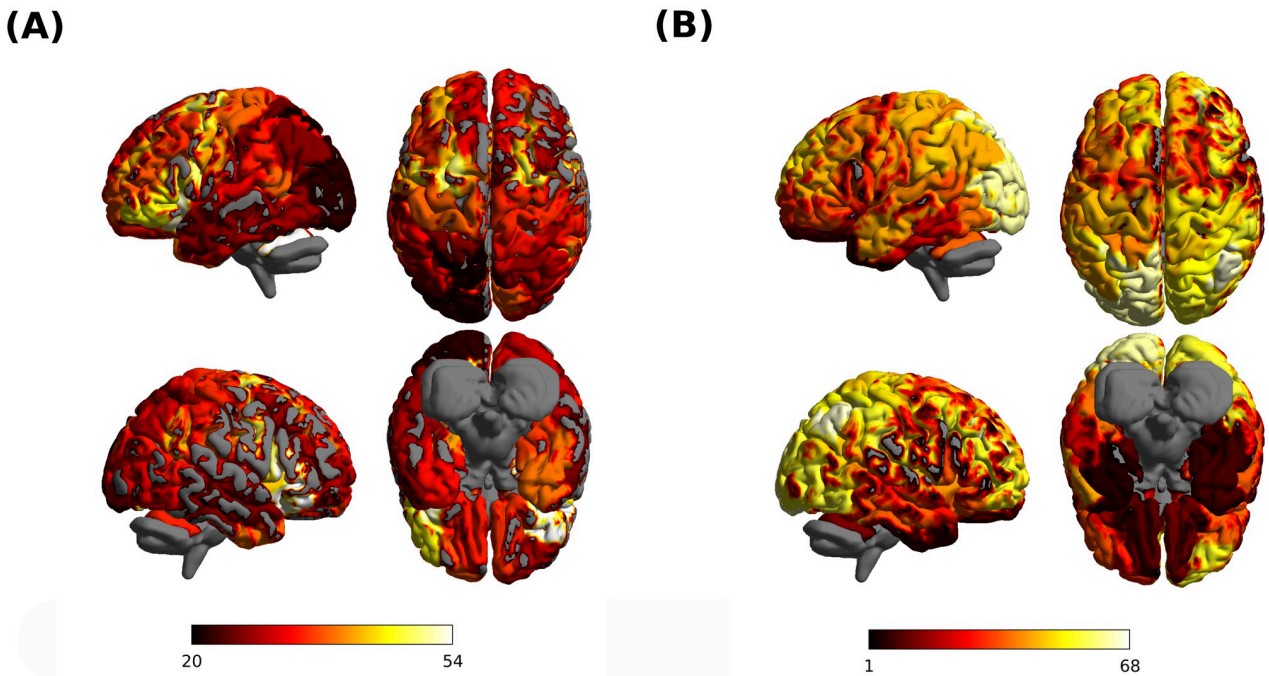

20                    54

1                    68

**Fig 3. Driver nodes and target nodes. (A)** We show a rendering of the brain, with each dot representing the center of one of the 74 regions. Node size is inversely proportional to the node driver centrality $\mathcal{E}_i^d$, while nodes color corresponds to resting state network affiliation. **(B)** as in (A), but node size is inversely proportional to the node target centrality $\mathcal{E}_i^t$. Brain images were visualized using BrainNetViewer [48].

The average $\mathcal{E}_i^t$ tends to decrease along the posterior/anterior axis, with posterior nodes generally corresponding to larger target energies (the correlation between $\mathcal{E}_i^t$ and the sagittal coordinate $y$ of the nodes is significant, $R = -0.34$, $p = 0.003$). Nodes with high rank (low $\mathcal{E}_i^d$), on average over subjects, include portions of the anterior DMN (ventrolateral prefrontal cortex/ba47 and dorsomedial prefrontal cortex/ba8), the anterior portion of the VAN (dorsolateral prefrontal cortex), the anterior portion of DAN (frontal eye field), primary motor cortex, putamen, left cerebellum, right hippocampus. Nodes with high rank (high $\mathcal{E}_i^d$) include thalamus, caudate, the temporal portion of DMN, primary visual cortex, the posterior portion of the DAN. However, in agreement with the above states caveat, the distribution (over subjects) of node ranks is quite broad for all nodes, with a st. dev. of $\approx 20$ for all nodes, implying that ranks are not consistent across subject (S5(A) Fig). Therefore, while we can identify nodes that tend to be better/worse as driver nodes across subjects, no nodes are consistently good/bad for all subjects. S5(A) Fig also shows node affiliation to one of eight resting state networks (RSN). The node ranking does not clearly correlate with RSNs affiliation: no networks are consistently associated with low/high ranks.

To identify nodes that are easy to control, we ranked nodes based on $\mathcal{E}_i^t$. Fig 3B shows a rendering of the brain, with region color corresponding to the average rank. The average $\mathcal{E}_i^t$ tends to decrease along the ventral/dorsal axis, and to increase along the posterior/anterior axis, with ventral and anterior nodes generally corresponding to larger target energies (we found a significant correlation between $\mathcal{E}_i^t$ and the axial coordinate $z$ of the nodes, $R = -0.63$, $p = 4 \cdot 10^{-9}$, and a significant correlation between $\mathcal{E}_i^t$ and the sagittal coordinate $y$ of the nodes, $R = -0.32$, $p = 0.005$). Nodes with low $\mathcal{E}_i^t$ include primary visual cortex, posterior nodes of the DAN,

posterior nodes of the CON, right anterior nodes of the CON, and the medial prefrontal cortex portion of the DMN. In terms of RSN affiliation, nodes of the Limbic network and subcortical nodes are generally associated with very low ranks. On the contrary, nodes of the control and the sensorimotor network are generally associated to high ranks. The distribution (across subjects) of node ranks, for each node, is shown in S5(B) Fig. In agreement with the above discussion of consistency, the rank distribution is much sharper than that obtained with $\mathcal{E}_i^d$, with a st. dev. of $< 10$ for many nodes. Therefore, not only can we identify nodes that tend to be better/worse as target nodes (in terms of control energy) across subjects, but we find nodes that are consistently good/bad for all subjects.

*In summary, in-hubs of EC are easy to control, out-hubs of EC are the best nodes to use as driver nodes; In-hubs are consistent over subjects, and generally located dorsally; out-hubs are poorly consistent over subjects, and mostly locate frontally.*

## RSN target controllability

We have shown that the control energy needed to control a target node depends on the choice of the driver nodes, and we linked this variability to the structure of effective couplings. Here, we address the general problem of selecting driver nodes when wishing to control more than one target nodes. Due to the general findings in the "scaling" subsection, we consider only small sets of target nodes. A natural choice is to take groups of nodes belonging to the same resting state network (RSNs) as targets. RSNs correspond to integrated neurocognitive systems [59–62] and are jointly affected in major brain disorders [30, 31].

**Driver node selection.**   We computed the control energy required to control each of eight RSNs, using a fixed number of driver nodes $n_d$. We systematically analyzed the effect of driver node selection, by comparing results obtained selecting driver nodes: i) based on a driver energy rank ii) based on EC centrality iii) randomly. Nodes belonging to the target RSN were excluded from the set of possible driver nodes. Results for $n_d = 5$ are shown in Fig 4, where we show the average (log)energy to control each RSN with different driver node selection.

For each RSN, energies were z-scored to the mean energy (over subjects and driver node selection). Performing a two-way repeated measures ANOVA on the z-scored energy values, with RSN and driver nodes selection criterion as factors, we obtained a significant effect of selection criterion ($F(5, 375) = 195, p < 10^{-10}$), and a significant criterion × RSN interaction ($F(35, 2625) = 5.2, p < 10^{-10}$). Post-hoc T-tests comparing different criteria to select control nodes show that selecting nodes based on driver centrality or EC centralities except page-rank (pq, ratio degree and out degree centralities) systematically yields lower energies than random ($T(75) < -21.6, p < 10^{-10}$). The strongest effects of node selection are felt in the small networks (LIM, VIS and SMN networks). For each target RSN, we identified the centrality yielding the lowest energy ("optimal centrality"), shown in Table 1.

**Energy to control a target RSN.**   We computed the control energy required to control each of eight RSNs, using a fixed number of driver nodes $n_d$. Nodes were selected according to the optimal centrality. In S6(A) Fig we show how the energy scales as a function of $n_d$. For $n_d = 2$, energies are $> 10^3$ for all RSNs ($10^3 - 10^8$). For $n_d = 5$, energies are in the range $10^{2.5} - 10^{4.5}$. With $n_d = 10$, energies are in the range $10^2 - 10^{3.5}$. Thus, 5 to 10 driver nodes would generally be required to control a target RSN with moderate energy. In S6(B) Fig we show the average (log-)energy, normalized by the number of target nodes included in each network. This allows a more direct comparison between different RSNs. It is clear that the DMN, DAN, CON, VAN and SUB are comparatively easier to control, while LIM, VIS and SMN are more difficult to control.

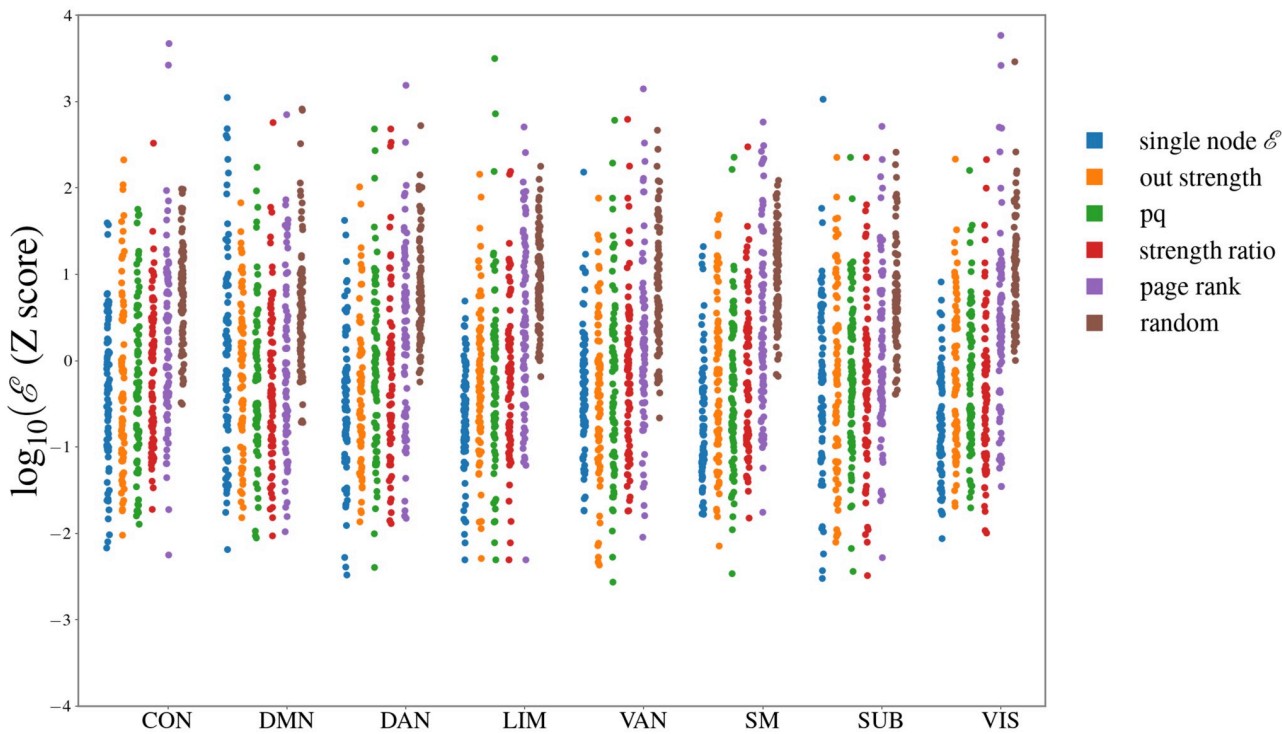

**Fig 4. Target RSN controllability.** For each of the eight RSNs, we computed the control energy required to control all nodes belonging to that RSN, when $n_d$ = 5 driver nodes are selected among all remote nodes (*not* belonging to that RSN) according to rankings based on centrality measures. For all RSNs and all choices of driver nodes, we plot the distribution over subjects of the average (log) energy per node.

**Controlling a RSN from another RSN.**   For each target RSN, we ranked nodes based on the optimal centrality and identified a set of 'optimal driver nodes' $n_d$ = 10 nodes with highest rank (at the individual level). We asked whether the optimal nodes to control a given target RSN preferentially belong to specific driver RSNs. In Fig 5A we show, for each target RSN, the average percentage of optimal nodes belonging to each driver RSN.

For each pair driver RSN/target RSN, we tested whether this fraction was higher or lower than expected randomly. Intuitively, if optimal nodes were selected randomly from any driver RSN, the fraction of optimal nodes from a given RSN should approximately match the fraction of nodes belonging to that RSN. More formally, a Fisher exact test (Methods) can be performed to identify when the fraction of nodes from a given driver RSN is lower/higher than

**Table 1. RSN Target control energy.**

| region | centrality | $n_t$ | $n_d$ | $\langle \log_{10}(\mathcal{E}) \rangle$ | $\langle \log_{10}(\mathcal{E}) \rangle_{agg}$ | $\Delta\mathcal{E}$ |
|---|---|---:|---:|---:|---:|---:|
| CON | out strength | 10 | 10 | 2.446 | 2.536 | 0.090 |
| DMN | out strength | 16 | 10 | 3.521 | 3.698 | 0.177 |
| DAN | single-node | 9 | 10 | 2.602 | 2.677 | 0.075 |
| LIM | single-node | 5 | 10 | 2.145 | 2.350 | 0.205 |
| VAN | out strength | 11 | 10 | 2.797 | 2.992 | 0.195 |
| SMN | single-node | 6 | 10 | 2.608 | 2.760 | 0.152 |
| SUB | pq | 12 | 10 | 3.298 | 3.472 | 0.174 |
| VIS | single-node | 5 | 10 | 2.252 | 2.401 | 0.151 |

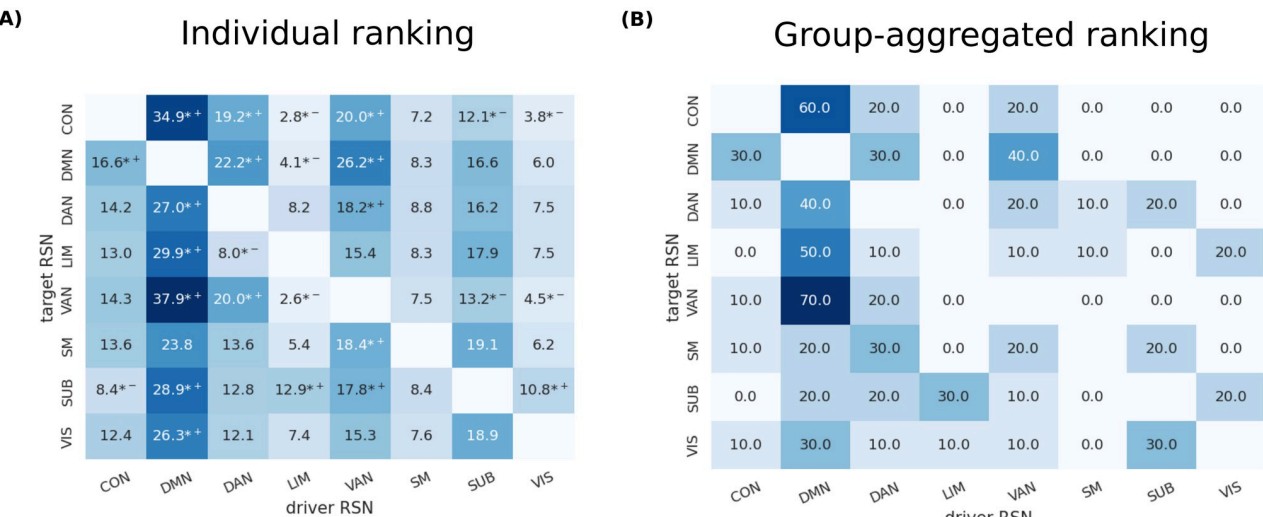

**Fig 5. Relation between driver and target RSNs.** For each target RSN and each subject, we obtained a ranking of driver nodes based on the centrality measure yielding the least average energy to control the target network. **A** For each RSN and subject, we considered the top 10 driver nodes according to the ranking. We plot the number of the top 10 driver nodes belonging to each RSN, on average over subjects. We mark with *+, *− nodes that significantly more/less represented among the top 10 driver nodes than expected by chance (Fisher exact test, $P < 0.05$, false-discovery-rate corrected for multiple comparisons (as we have 8 RSNs, we have $8 \times 8 - 8 = 56$ comparisons, as we consider all possible pairs of driver RSN—target RSN excluding identical pairs). **B** Using rank aggregation, for each target RSN we obtained a single ranking for all subjects and considered the top 10 driver nodes. We plot the number of the top 10 driver nodes belonging to each RSN.

chance (marked with *+, *− in Fig 5a). Notably, the DMN is overrepresented among good drivers of nearly all networks. The VAN and DAN are overrepresented in the control of each other and the DMN and CON. Conversely, the LIM is systematically underrepresented.

**Individual vs. group selection of driver nodes.** For each target RSN, we used rank aggregation to combine individual rankings in a unique ranking (Methods), obtaining a group-wise set of optimal driver nodes. In Fig 5B we show, for each target RSN, the average percentage of group-optimal nodes belonging to each driver RSN. Results are very similar to Fig 5A, but rank aggregation tends to sparsify the matrix. We observe that the SMN is very underrepresented among top-ranking driver nodes. This is probably a consequence of the fact that effective connections of the SMN are highly variable among subjects, so that no nodes of the SMN consistently appear among the top-ranking for many subjects. In Fig 6 we show the 10 top-ranking nodes according to the aggregated ranking, for two example target RSNs (CON and SUB). In S7 Fig we show results for all target RSNs.

Among the nodes frequently represented we find: the ventrolateral prefrontal cortex nodes of the DMN and VAN (which among the top-10 ranking nodes for nearly all target networks), the frontal nodes of the DAN, the dorsomedial nodes of the DMN, the precuneus, the striatum and the left cerebellum (for a more detailed discussion, see S2 Text). We asked to what extent group results, i.e., the aggregated ranking, can be used to select driver nodes. Therefore, we compared the energy to control each target RSN, averaged over subjects, when nodes were selected based on an individual node ranking ($\langle \log_{10} \mathcal{E} \rangle$) or the aggregated ranking $\langle \log_{10} \mathcal{E} \rangle_{agg}$. Results are shown in Table 1. Obviously, the individual ranking is more efficient ($\Delta \mathcal{E} = \langle \log_{10} \mathcal{E} \rangle - \langle \log_{10} \mathcal{E} \rangle_{agg} > 0$). However, the difference is small, ranging from a $\Delta \mathcal{E} = 0.075$ for DAN (corresponding to a factor 1.3 in energy) to $\Delta \mathcal{E} = 0.205$ for VAN (corresponding to a factor 1.6 in energy). Therefore, the group results can be used to inform the node selection. We also verified that selecting driver nodes based on centrality measures computed on

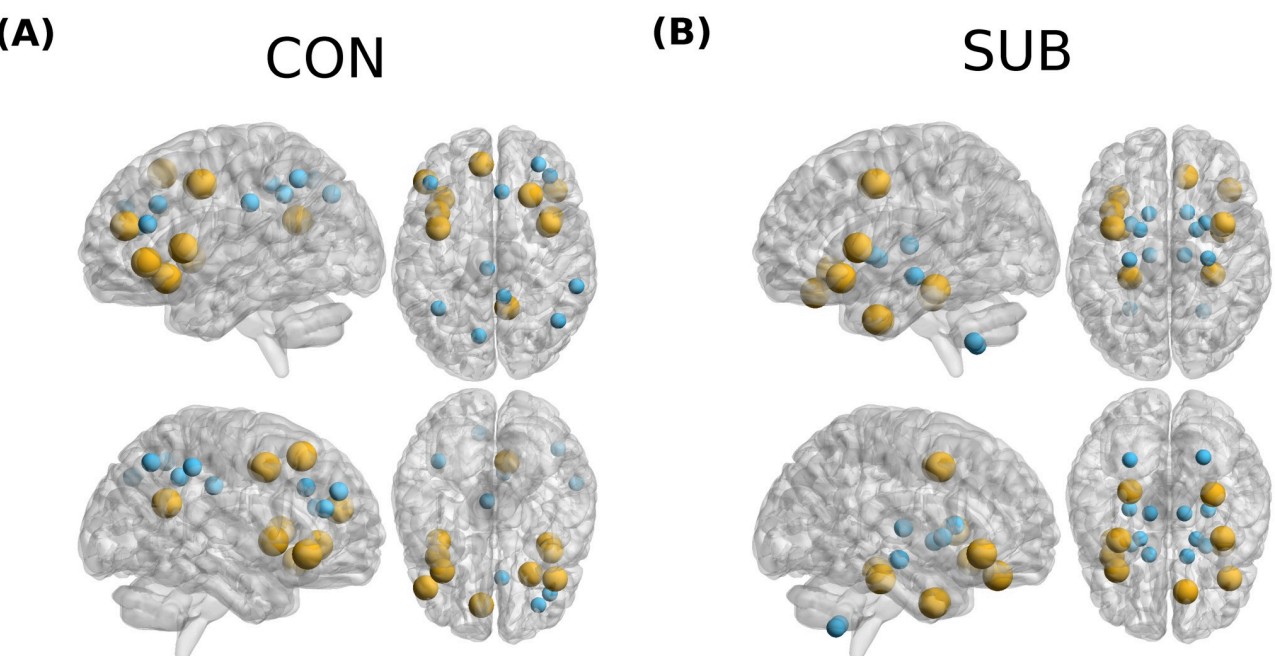

**Fig 6. Optimal driver nodes for RSN).** Two example RSNs with the corresponding target nodes(small blue markers) and top 10 aggregate driver nodes (yellow markers) are shown. Brain images were visualized using BrainNetViewer [48].

the FC (instead of the EC) leads to larger control energies. Results are shown in S1 Table, where we compare three quantities: i) the average difference in (log-)energies when using group ranking vs individual ranking, based on FC. Since FC estimates are less variable over subjects than EC estimates, the difference is generally smaller than the difference obtained using an EC-based ranking. ii) the average difference in (log-)energies when using individual ranking, based on FC vs EC. The difference is always positive. iii) the average difference in (log-)energies when using group ranking, based on FC vs EC. The difference is always positive, except for DMN. Thus, while using FC to select centrality measures allows a greater robustness of group results, control energies are still larger than those obtained when using EC-based centrality measures. Notably, when using FC, the optimal centrality criterion is nearly always the page rank centrality, implying that optimal control nodes are generally those that *do not* receive many incoming connections (hence, not FC hubs).

## Discussion

### Effective-connectivity-based controllability

We proposed an approach to brain controllability based on effective connectivity (EC) inferred from fMRI, instead of structural connectivity (SC) as in the standard approach. To what extent EC depends on the underlying SC is an open question [63]. The EC model is in principle better suited to represent activity propagation, but we are not aware of previous studies presenting a thorough analysis of controllability properties of whole-brain EC networks. Ref. [17] computed controllability metrics on EC networks, but the entire analysis focused on a small subset regions involved in cognitive control. Recent work [64, 65] assessed controllability properties of functional rather than structural networks, but used standard functional connectivity (FC) networks, rather than effective connectivity (EC) networks. We stress that standard FC only

measures correlations between the signals of two areas, and in principle it is not suited to model activity propagation. A key trait of EC networks is that they are asymmetric, contrary to FC networks (symmetric by definition) and SC networks (symmetric due to limitations of diffusion MRI). Consistently with other works [23], we observed substantial asymmetries in EC connections. We showed that asymmetries are important for control, as incoming and outgoing connections, and correspondingly in- and out-hubs, play a different role (S3(A) Fig). This differences cannot be detected with an SC- or FC-based analysis. In future work, it is certainly of interest to compare SC- and EC-based predictions of activity propagation following neuromodulation in a TMS-EEG experiment (a recent study [66] compared SC- and FC- based prediction, but FC cannot adequately predict propagation patterns). To ensure that our EC-based scheme is not sensitive to small errors in EC estimates, we performed a sensitivity analysis. For each subject independently, we added random Gaussian noise so as to perturb each effective connection with a relative error $f$ up to 25%. For each value of $f$, we assessed the fraction of top-10 ranking nodes (according to several centrality measures) shared between the unperturbed and perturbed case. Results are shown in S10 Fig. The set of optimal nodes is very robust with respect to small variations of the effective connectivity estimates. On average (over subjects) the fraction of top-ranking nodes stays above 90% (for out strength), above 85% (for ratio degree centrality), above 40% (for pq centrality). The pq centrality is more fragile, because adding Gaussian noise has the effect of perturbing the spectral properties of $A$, creating large eigenvalues which strongly distort the spectral properties of the Gramian.

## Impracticability of whole-brain control

Based on Kalman's criterion, Gu et al. [5] originally argued that the whole brain could be controlled by acting on a single driver node. However, Kalman's criterion does not ensure practical controllability. The latter requires a reasonably low power of the control signal, i.e., a reasonably low control energy. Very large values of the control energy are problematic for the following reasons: i) Experimentally, control signals are subjected to bandwidth and power constraints; ii) Control trajectories become highly non-local and extremely long, and, consequently, numerically unstable: small numerical errors in the Gramian $W$ (of magnitude comparable to machine precision) imply very large deviations in the final state, and thus it is not practiaclly possible to reach the desired target [21]; iii) Numerical instabilities can be exacerbated by noise and model inaccuracies. Focusing on SC controllability, Tu et al. [6] showed that control energies are astronomically large unless an important fraction of the network nodes ($\gtrsim 20\%$) are used as driver nodes. Here, we replicate Tu et al.'s findings with EC controllability, showing that whole-brain control implies exceedingly high control energies unless 15–20% of the nodes are used as driver nodes (Fig 1A). In fact, huge control energies should be expected whenever the ratio between number of driver nodes and the number of target nodes is small, irrespective of details of the system or model at hand. Indeed, the control energy scales (roughly) exponentially with the number of target nodes to be controlled (Fig 1C).

## Target controllability of brain networks

Currently available techniques for non-invasive brain stimulation, such as transcranial magnetic stimulation (TMS), do not allow for stimulation of multiple sites. Standard TMS allows stimulation of one site, and recent experiments with dual coil TMS stimulation (also known as cortico-cortical paired associative stimulation [22, 67]) allow stimulating two cortical sites. Proposals to implement multi-site stimulation are currently under development [68], but formidable technical difficulties must be surpassed before stimulation of more than a few sites

simultaneously becomes possible. Therefore, we decided to focus our analysis on control of one or a few target nodes, adopting the framework of target controllability [27, 69]. While recent work considered target controllability of whole-brain brain networks [70, 71], we are not aware of a systematic study of target controllability on EC networks. We first analysed the case where a single driver node is used to control a single target nodes (corresponding to experiments where one tries to activate a chosen brain region by acting on a remote region). Controlling a single target region is feasible with limited control energy ($\lesssim 10^1$). We then considered the case of controlling a subset of nodes belonging to the same resting state network (RSN). Control of target RSNs demands large control energies unless at least $n_d \approx 5$ driver nodes are used. Therefore, the control objective is not easily accomplished with current technology. We stress, however, that controlling a subnetwork (in the sense of target controllability) is equivalent to being able to generate an arbitrary activity pattern in the subnetwork. This is considerably more difficult than producing a generic overall activation of the subnetwork (as in [72]).

## Criteria for selection of driver nodes

When considering targets of small size, it is relevant to properly select driver nodes, ensuring a low control energy. A random selection of nodes is generally inefficient (Fig 1). In the single-driver-single-target case, control energies span 3 orders of magnitude depending on which node is selected (Fig 2A). Good driver nodes should be connected to the target: owing to network effects, both direct and indirect connections count (Fig 2A and 2B), such that there is a general relation between control energy and network distance between driver and target (Fig 2C). Negative (inhibitory) connections appear to have the same effectiveness for control as positive (excitatory) connections (Fig 2E). To identified node that are generally "good drivers" or "good targets", we defined two control metrics, a *driver centrality* and *target centrality*, averaging, respectively, the energy required to control other nodes of the network from a given driver node and the energy required to control a given target node from other nodes. The driver and target centralities are strongly related to EC graph centralities. In particular, out-hubs of EC (nodes with a large weight of outgoing connections) serve as good control drivers, while in-hubs of EC (nodes with a large weight of incoming connections) are easy target nodes. We thus replicate qualitatively a major finding of previous studies on brain controllability [1, 5], namely, that hubs correspond to optimal driver nodes. However, within our picture there is a clear distinction between in-hubs and out-hubs—a difference that can be appreciated only when considering asymmetric effective connections. When using several driver nodes (e.g., to control a target RSN), it is impossible to perform an exhaustive search of the optimal subset driver nodes, but node centralities can be used to choose the drivers. Selecting driver nodes from a ranking based on driver centrality or EC graph centralities (pq, ratio-degree or out-strength centrality) yields a significant energy advantage over a random selection of driver nodes (Fig 4). All these metrics largely correlate with the out-strength centrality. We stress that the relevance of out-hubs for control, i.e., to transmit external perturbations to distal nodes, does not imply that these nodes are the most relevant or influential in spontaneous brain dynamics. In particular, if one ranks nodes on the basis of the impact their *removal* can have on brain dynamics [73], then the most relevant nodes may correspond to other topological criteria, such as belonging to the set of rich-club nodes.

## Node accessibility

Throughout this study, we have assumed that driver nodes can be selected freely. However, depending on the stimulation method chosen (TMS, TACS, DBS etc.), not all nodes might be

equally accessible to drive the network. In practice, this means that the control signal cannot reach the nodes, or it can reach it with reduced power. We might define a *node accessibility* $\beta_i$, with $0 \leq \beta_i \leq 1$, such that any control signal $u(t)$ acting node $i$ is reduced by $\beta_i$: $\beta_i = 1$ corresponds to fully accessible nodes, $\beta_i = 0$ corresponds to fully inaccessible ones. This is equivalent to modulating the entries of the matrix $B$ in Eq (8), by taking $B_{ij} = \beta_i$ instead of $B_{ij} = 1$ for all nonzero entries of $B$.

In the presence on non-unit values of the accessibility, centrality measures defined only on the effective connectivity matrix $A$ will not lead to a good driver nodes selection (as they do not take node accessibility into account). Conversely, the pq-centrality $r_i$(Methods) is based on the controllability Gramian, which takes into account also the matrix $B$. With non-unit values of the accessibility, we have $r_i = \beta_i^2 r_i^0$, where $r_i^0$ is the value obtained in the case where all nodes are fully accessible. In other words, the pq-centrality of node $i$ scales with node accessibility as $\beta_i^2$. Thus, this centrality measure accounts for both node topology and node accessibility. We simulated a situation where nodes had very different accessibilities. We generated random accessibilities between 0.01 and 1, mimicking a situation where some nodes are nearly inaccessible and some are fully accessible. In S8 Fig we show the control energy as a function of the number of driver nodes for different driver selection criteria. The lowest energy is obtained using pq-centrality, while the random selection is very inefficient (not surprisingly, as a random selection will generally include low-accessibility nodes).

## Individual vs. group selection of drivers

While in-hubs of EC were consistent over subjects, out-hubs exhibited a much larger inter-subject variability. Therefore, the location of a single "optimal driver node" may strongly vary among different individuals. This finding strengthens the case for an individual optimization of the stimulation targets, as advocated by recent contributions [72, 74]. However, when using multiple driver nodes, selecting nodes based on a group-averaged node ranking does not entail a significant additional energy cost with respect to using a subject-dependent ranking. Possibly, the usage of multiple nodes may partially offset fine-grained individual differences in the connectome profiles. This result is interesting in the light of the development of multiple-site neurostimulation paradigms: it implies that using a standardized protocol over different subjects, certainly convenient especially in a development stage, may not determine severe efficiency trade-offs.

## Optimal driver regions

Nodes with low driver centrality (good drivers) are predominantly located frontally (Fig 3A). This finding is in agreement with a previously reported anterior-to-posterior information flow in the slow frequency range [75]. Good drivers include dorsolateral and ventrolateral prefrontal nodes of the CON, DAN and DMN, primary motor areas, and left cerebellum. The driver centrality, being an average measure, is not sensitive to different targets. By using rank aggregation, we found nodes that are frequently ranked among good driver nodes for several target RSNs. The most recurring nodes are DAN and DMN nodes located in the ventrolateral prefrontal cortex, the frontal eye field (DAN), and hubs of the DMN (dorsomedial nodes of the DMN and precuneus), the cerebellum and the striatum. Dorsolateral and ventrolateral prefrontal regions are among the key regions involved in cognitive control [76–81]. The frontal eye field is a key region mediating attentional control [82]. Primary motor regions, that are involved not only in motor control but also a wide array of top-down processes [83, 84], have been previously associated to a high control centrality [5, 71]. The left cerebellum is strongly involved in motor control of the dominant hemisphere [85]. The striatum is widely implicated in learning and reward [86]. The topography of good driver nodes generally aligns with the

cortical hierarchy [87]. Areas from attentional and association networks high in the cortical hierarchy (DAN, DMN, VAN) are consistently identified as good driver nodes, contrary to areas low in the cortical hierarchy (including somatomotor, visual and limbic areas). This trend culminates with the DMN, which sits on top of the cortical hierarchy and is overrepresented among driver nodes of all target networks, coherently with a hypothesized central integrative role of the DMN in the brain [88]. We note that these findings do not fully align with those of Ref. [71], which indicated only motor regions as optimal driver regions. However, Ref. [71] based the controllability analysis on a "directed structural connectome" obtained by normalizing outgoing connections by node degree, assuming a diffusive process on the network [89]. The ensuing network considerably differs from EC network empirically observed in fMRI, and implies robust outgoing connections for nodes with low degree, which are mainly located in the somatomotor cortex.

## Optimal targets

Target centrality is organized along the axial direction, with ventral nodes being generally associated with higher target centrality (Fig 3B). In particular, subcortical nodes and nodes of the limbic network correspond to particularly large control energies, implying that they are much less easy to activate and control from remote regions, in agreement with previous findings [74]. Among the nodes with lowest target centrality (hence, among the nodes that are most easy to perturb remotely) are frontoparietal nodes belonging to the control network, which integrate input from several regions and plays a central role in cognitive control [77].

## Limitations

We finally discuss possible limitations of the present work. *Cohort and recordings.* The sample used ($N = 76$) was not large enough to split our analysis into a training and a validation cohort. Furthermore, the relatively short fMRI time-series (here, 657 time points) are not optimal for the stability of individual-level EC matrices. Thus, a replication in a different data set involving a larger cohort and longer recordings may considerably strengthen our analysis. A larger cohort would possibly allow linking control properties (such as driver and target energies) with individual traits, e.g., demographic data, allowing to look for age-, sex- and parenthood-related effects ([71, 90]). *Parcellation.* The general findings of our manuscript do not depend on the specific parcellation used, but some of the more specific findings may not be parcellation-invariant. For instance, while the relation between out-degree and driver centrality is expected to be general, the specific identity of the optimal driver nodes may depend on the parcellation, as the in- and out-degree of different nodes may slightly vary in different parcellations. We also warn the reader that the specific values of control energy obtained depend on the parcellation used: more fine-grained parcellations imply a larger number of nodes, and hence an increased difficulty of the control problem. *Control cost.* In the present work we used the control energy as general measure of control cost. Control energy is a worst-case-scenario metric, as it measures the maximal square amplitude of the control signal required to generate a desired activity pattern in the target region (maximum over all possible patterns). Therefore, our estimates of controllability are generally quite conservative. *Controllability framework.* Here, we used the standard framework of linear controllability. Obviously, a limitation of this framework is the assumption that the dynamics is linear. However, we acknowledge two additional, potentially more relevant limitations. First, the framework is not fault-tolerant, as it neglects noise and aims at inducing an exact target pattern. In a recent publication, Kamiya et al. [91] framed the control problem probabilistically: the system's state is not a specific activity pattern, but a distribution, and the control objective is to reach a target distribution

centered around a specific pattern. In principle, one could use this approach in combination with sparse DCM to achieve a fault-tolerant control approach, but this would require some advances: in its current formulation, Kamiya et al's approach assumes that one can control all network nodes, and we should therefore adapt it to embed stronger constraints on the driver nodes that can be used. A second limitation is the framework allows for arbitrary control signals *u*: in practice, there are often constraints in the control signals that can be generated, e.g., in terms of bandwidth. Finally, in terms of applications, the control approach we are using is suitable for a "single shot" application where one temporarily induces a desired activity state. For clinical applications, it would be relevant to understand how repeated stimulation can leverage plasticity mechanisms inducing long-term changes [92]. *Rapid effects of brain stimulation*. EC inferred from fMRI is represents infraslow activity (which mostly represents slow modulations of gamma activity [93]), and hence it can capture a slow activity propagation, on the order of several seconds, originating from a node whose local activity is increased by stimulation (see S2 Fig). It cannot, however, account for a rapid non-local effect of neurostimulation, which induces local spiking and hence also a fast activity spread along anatomical pathways [94]. How to model this effect remains an open challenge. *Task experiments*. The dynamical model, Eq (1), and the EC matrix were derived for the resting state condition. As such, our control model is appropriate only during rest. Note that several works already used fMRI-based connectivity results (such as RSNs) to select driver nodes for TMS-EEG experiments [22, 95, 96]. During a tasks, we cannot generally model the dynamics as linear time-invariant (LTI). In fact, the dynamics can contain task-dependent inputs modulating the regions' activity and effective connections [97], leading to a non-linear and non-time-invariant dynamics. To what extent task-dependent modulations can be neglected, using a single task-related effective connectivity matrix, and to what extent the optimal control nodes identified at rest would still be efficient during a task are open questions that would deserve a dedicated study. *Whole-brain computational model*. The DCM model used in this study can yield a very accurate estimate of the effective connectivity, but is dynamically simplistic. To what extent predictions of our linear model would be fully accurate in the case of a more complex, non-linear node dynamics is an open question. In future work, we will address this question by testing our predictions on simulations of several whole-brain model with non-trivial node dynamics. To this aim, we will use models including a data-driven, asymmetric EC estimate (the key feature for our approach) and non-trivial node dynamics, such as the model proposed by Deco et al. [98].

## Supporting information

**S1 Text. Linear controllability.** Details on the derivation of the main linear controlability formulas reported in Methods.
(PDF)

**S2 Text. Optimal nodes to control RSNs according to rank aggregation.** Details optimal nodes to control each target RSN.
(PDF)

**S1 Fig. Examples of effective connectivity matrices.** The 74 brain areas are divided in left cortical, right cortical and subcortical areas. The figure in panel (C) represents the asymmetry in $A$, defined as $\delta A = A - A^T$.
(PDF)

**S2 Fig. Functional propagation of perturbations.** We simulated the propagation of a perturbation in the model $\dot{\mathbf{x}}(t) = A\mathbf{x}(t)$ where $A$ is the average (over subjects) effective connectivity matrix. The system's state was initialized to $\mathbf{x}(0) = 0$ (corresponding to the stable equilibrium

point) except for the state of node $i$ that was perturbed to $x_i(0) = 1$. We let the system evolve freely until $t = 100$. The signal of area $j$ is the functional response of area $j$ to the perturbation in $i$. **(A)** functional in all areas $j \neq 0$ when node $i = 0$ is perturbed. **(B)** For each region (except the pertubed region $i$), we identified the time corresponding to the maximum of the functional response. We iterated this procedure perturbing all $n$ regions in the network, obtaining $n \cdot (n-1)$) response peak times. We show the histogram of peak response times. The average response time is $T = 10.29$.
(PDF)

**S3 Fig. Relation of driver and target control energy and effective connectivity. (A)** For each subject, we computed the Pearson correlation $R$ between the node driver and target centrality $\mathcal{E}_i^d$, $\mathcal{E}_i^t$ (average energy to control other nodes from node $i$ vs average energy to control node $i$ from other nodes) and the in-strength $A_i^{in}$ and out-strength $A_i^{out}$ of effective connections, as well as the strength of functional connections $\mathcal{F}_i$. We show the distribution of $|R|$ over subjects. **(B)** For each node, we computed the coefficient of variation (s.d./mean) over subjects of $\mathcal{E}_i^d$, $\mathcal{E}_i^t$, $A_i^{out}$ $A_i^{in}$. We show the distribution of the coefficient of variation over nodes.
(PDF)

**S4 Fig. In-hubs and Out-hubs. (A)** Difference between in-hubs and out-hubs **(B)** Hubs of the functional connectivity (in-hubs and out-hubs coincide because the functional connectivity is symmetric). Brain regions are colored according to the FC strength (averaged over subjects) **(C)** Out-hubs of the effective connectivity. Brain regions are colored according to the EC out-strength (averaged over subjects) **(D)** In-hubs of the effective connectivity. Brain regions are colored according to the EC in-strength (averaged over subjects). Brain images were visualized using BrainNetViewer (Xia M, Wang J, He Y. BrainNet Viewer: a network visualization tool for human brain connectomics. PloS one. 2013;8(7):e68910).
(PDF)

**S5 Fig. Node ranks based on driver/target control energy. (A)** For each subject, we ranked all nodes based on the value of driver centrality $\mathcal{E}_i^d$. We show the rank distribution for all nodes, with nodes ordered according to the average rank, from lowest to highest. Nodes are colored according to the resting state network they belong to. **(B)** Same as (A), but ranks are based on target centrality $\mathcal{E}_i^t$.
(PDF)

**S6 Fig. Dependence of the RSN control energy on the number of driver nodes.** Energy to control RSN with a varying number of driver nodes, selecting driver nodes according to the driver centrality **(A)** Energy to control RSN with a varying number of driver nodes, rescaled by the number of target nodes $n$.
(PDF)

**S7 Fig. Optimal driver nodes for RSN.** For each RSN we show the corresponding target nodes(small blue markers) and top 10 aggregate driver nodes(yellow markers) are shown. Brain images were visualized using BrainNetViewer (Xia M, Wang J, He Y. BrainNet Viewer: a network visualization tool for human brain connectomics. PloS one. 2013;8(7):e68910).
(PDF)

**S8 Fig. Control energy in presence of non-unit node accessibilities. (A)** Energy to control the whole brain network (median over subjects) as a function of the number of driver nodes $n_d$. For each $n_d$, nodes were selected based on a ranking of centrality measures. Nodes had different accessibility values $\beta_i$ in the range [0.01, 1]. **(B)** Energy to control the whole brain

network (distribution over subjects), for three values of $n_d$. For each subject, energy values were z-scored with respect to the mean of the random node selection.
(PDF)

**S9 Fig. Selecting nodes based on functional connectivity rather than effective connectivity.** **(A)** Fraction of common nodes (distribution over subjects) among the top-10 ranking nodes according to different centrality measures based on FC and EC. **(B)** Energy to control target nodes, using $n_d$ = 10 driver nodes (distribution over subjects) selected according to different centrality measures. Centrality measures were computed on FC instead of EC. For each number of target nodes, energy values were z-scored with respect to the mean energy obtained with the same centrality but using EC.
(PDF)

**S10 Fig. Effect of perturbing the effective connectivity on driver node selection.** For each subject, we considered the effective connectivity matrix $A$. We perturbed $A$ by adding random Gaussian noise to each connection. The noise was chosen to have mean 0 and standard deviation proportional to the connection value, i.e., the noise $\epsilon_{ij}$ acting on $A_{ji}$ was $\epsilon_{ij} \sim \mathcal{N}(0, f \cdot A_{ji})$ with $0 \leq f \leq 0.25$. Thus, each link was perturbed by a relative error with magnitude $f$ with $f$ up to 25%. For each value of $f$, we ranked nodes according to several centrality measures and we assessed the fraction of top-10 ranking nodes common between the unperturbed ($f = 0$) and perturbed case. **(A)** Fraction of common nodes (distribution over subjects) among the top-10 ranking nodes according to different centrality measures based on FC and EC. **(B)** Energy to control target nodes, using $n_d$ = 10 driver nodes (distribution over subjects) selected according to different centrality measures. Centrality measures were computed on FC instead of EC. For each number of target nodes, energy values were z-scored with respect to the mean energy obtained with the same centrality but using EC.
(PDF)

**S1 Table. RSN Target control with functional-connectivity-based node selection.** We repeated the analysis of control energies for RSN targets, selecting the driver nodes on the basis of centrality measures computed on FC. Here, we show three quantities: i) the average difference in (log-)energies when using group ranking vs individual ranking, based on FC: $\Delta \mathcal{E}_{FC_{agg}}^{FC} = \langle \log_{10} \mathcal{E} \rangle_{FC,agg} - \langle \log_{10} \mathcal{E} \rangle_{FC}$ ii) the average difference in (log-)energies when using individual ranking, based on FC vs EC: $\Delta \mathcal{E}_{EC}^{FC} = \langle \log_{10} \mathcal{E} \rangle_{FC} - \langle \log_{10} \mathcal{E} \rangle_{EC}$ iii) the average difference in (log-)energies when using group ranking, based on FC vs EC: $\Delta \mathcal{E}_{EC_{agg}}^{FC_{agg}} = \langle \log_{10} \mathcal{E} \rangle_{FC,agg} - \langle \log_{10} \mathcal{E} \rangle_{EC,agg}$.
(PDF)

## Author Contributions

**Conceptualization:** Karan Kabbur Hanumanthappa Manjunatha, Samir Suweis, Michele Allegra.

**Data curation:** Giorgia Baron.

**Formal analysis:** Karan Kabbur Hanumanthappa Manjunatha, Giorgia Baron, Danilo Benozzo, Michele Allegra.

**Funding acquisition:** Maurizio Corbetta, Alessandro Chiuso, Alessandra Bertoldo.

**Investigation:** Karan Kabbur Hanumanthappa Manjunatha, Giorgia Baron, Michele Allegra.

**Methodology:** Karan Kabbur Hanumanthappa Manjunatha, Giorgia Baron, Danilo Benozzo, Maurizio Corbetta, Alessandra Bertoldo, Samir Suweis, Michele Allegra.

**Project administration:** Alessandro Chiuso.

**Software:** Karan Kabbur Hanumanthappa Manjunatha, Michele Allegra.

**Supervision:** Erica Silvestri, Alessandra Bertoldo, Samir Suweis, Michele Allegra.

**Visualization:** Karan Kabbur Hanumanthappa Manjunatha, Michele Allegra.

**Writing – original draft:** Karan Kabbur Hanumanthappa Manjunatha, Michele Allegra.

**Writing – review & editing:** Giorgia Baron, Danilo Benozzo, Erica Silvestri, Maurizio Corbetta, Alessandra Bertoldo, Samir Suweis.

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
