## [Decision Letter · Decision Letter 0]

20 Sep 2023

Dear Dr Allegra,

Thank you very much for submitting your manuscript "Controlling target brain regions by optimal selection of input nodes" for consideration at PLOS Computational Biology. As with all papers reviewed by the journal, your manuscript was reviewed by members of the editorial board and by several independent reviewers. The reviewers appreciated the attention to an important topic. Based on the reviews, we are likely to accept this manuscript for publication, providing that you modify the manuscript according to the review recommendations.

Sincerely,

Hayriye Cagnan

Academic Editor

PLOS Computational Biology

Lyle Graham

Section Editor

PLOS Computational Biology

Reviewer's Responses to Questions

**Comments to the Authors:**

Reviewer #1: This is a valuable contribution to the brain stimulation literature and I have enjoyed reading this manuscript. Some concerns are summarised below:

Depending on the stimulation method chosen (TMS, TACS, DBS etc.), not all nodes would be accessible to drive/control the network. How can we take this into account within this framework? i.e. not all driver nodes, irrespective of centrality, would have the same utility when access is considered. Could the authors incorporate “ability to access a node with certain stimulation techniques” into their centrality measure? (invasive vs noninvasive stimulation techniques)

What does T=10 correspond to in terms of time (i.e. in seconds)? (line 338) Is this a relevant timescale for TMS-EEG studies?

While using EC over SC is an interesting approach, are the authors working under the assumption that RSNs are reflective of valid target-driver node definitions for task based TMS-EEG applications? If so, please include relevant experimental evidence as references. I appreciate the authors are somewhat touching upon this as a “limitation: rapid effects of brain stimulation” but I think the issue is not just constrained to propagation of stimulation effects but rather the definition of states that can be modelled as linear and time-invariant over time periods relevant for a stimulation study.

Could you please elaborate further the factors(s) contributing to the asymmetry in the definition of in-hubs vs out-hubs across subjects?

The authors focused primarily on the stimulation-based literature. How does their approach relate to other methods employed in epilepsy surgery: e.g.

https://journals.plos.org/ploscompbiol/article?id=10.1371/journal.pcbi.1005637

Minor comments: the authors should carefully review their spelling throughout the manuscript – an example sentence 667-672 however this issue is present throughout.

Reviewer #2: Manjunatha et al. present a manuscript introducing a novel methodology to estimate brain “controllability” from (asymmetric) estimates of effective connectivity (EC) estimated using sparse Dynamic Causal Modelling (sDCM). Using these directed graphs (fit to subject level resting state fMRI), the authors deploy Kalman’s controllability framework to identify optimal nodes as either “drivers” or “targets” as determined by a theoretical energy requirement dependent on graph properties alone. The authors provide convincing evidence that identified "in" and "out" hubs in the brain have distinct energy demands for control. The paper is precise and well written, and present a number of insightful findings. The paper is highly theoretical however, and lacks any analysis of sensitivity or validation of the method that would improve its impact. I have a few issues that I recommend be addressed:

General considerations

1. This paper introduces EC estimates to replace more standardly used functional connectivity estimates of graph connectivity. I understand that asymmetry allows for hubs to be better characterized, but it is not clear exactly what is lost/gained. For instance, what are the overlaps in nodes identified with FC vs EC approaches? I suspect also that the EC estimates introduces much more variability between subjects, so how does EC and FC compare when performing group level identification of optimal nodes.

2. How sensitive is the controllability analysis and following identification of node optimality affected by small variations in EC estimates? If we prune away or invert a particular connection, how much does node selection change? A perturbation analysis, taking into account the posterior precisions over connectivity parameters would be particularly interesting to this respect.

3. The paper would be more impactful if it contained an empirical validation to test the underlying assumptions of the approach. The introduction states current failures of controllability arise from “the usage of inaccurate computational models of brain dynamics”. This work deploys directed connectivity as its principle novelty – does this added description suffice to improve controllability? A set of numerical simulations of delayed, nonlinear models could help validate the approach by demonstrating that simplifying assumptions (such as ignoring node-dynamics or delays) can still yield prediction useful in real world systems. I understand this might be out the scope of current paper, so leave it to the authors as to how best to address this.

Note: Line numbering is not consistent across the manuscript – I have indicated lines nearest.

Minor Points:

Introduction

• To make accessible to more general audience – could you address how this stimulation approach would look in practice?

• The discussion of whole brain vs small node control is interesting, and seems well justified.

Methods

• L135-136 - “The algorithm has been further adjusted to account for the signal reliability of the temporal frames, by introducing the binary temporal mask as a weighting measure during the estimation procedure.” – its not clear what this means, could you please clarify?

• L201 - a textual description of the interpretation of driver and target centrality metrics follows later in the text, I think it would be clearer to move this up to this point.

Results

• Figures –axis labels use mathematical notation which makes them hard to read outside of the text. I would suggest to use natural language and be clear to indicate which colour legends apply to which figures.

• The intuition behind the explanation underlying the identification of in-hubs as optimal targets for stimulation. I would also expect that an element of controllability pertains to how much “competition” a node is receiving. If the driving node is competing with lots of other extraneous nodes for control then surely the energy will be higher?

• Figure 3 – I am not sure what these are meant to show. That good drivers are more frontally distributed? At the moment this isn’t entirely clear.

• Nodes with low rank L391 – should this be “high rank”?

• Can you normalize the energy by the RSN size? You’ve already shown smaller networks are going to be easier to control, so a normalized metric would be useful.

• Figure 5a – what is the reasons for doing these analyses? Do you hypothesize that one RSN drives another? This is not really considered in the introduction.

Discussion

• What are the biases inherent in estimating the EC? Do regions with a stronger BOLD signal tend to be demarked as in- or out- hubs? Can this explain why out-hubs have high variability, because they tend to be located in regions of low SNR?

• I suspect that the discussion relating to “optimal driver regions” is going to change depending on what type of data you have (i.e., resting or task based). Could you comment on how stable these driver optimalities are likely to be?

**Have the authors made all data and (if applicable) computational code underlying the findings in their manuscript fully available?**

Reviewer #1: None

Reviewer #2: Yes

PLOS authors have the option to publish the peer review history of their article (what does this mean?). If published, this will include your full peer review and any attached files.

Reviewer #1: No

Reviewer #2: No

Figure Files:

Data Requirements:

Reproducibility:

References:

---

## [Editor Report · Decision Letter 1]

4 Dec 2023

Dear Dr Michele Allegra,

We are pleased to inform you that your manuscript 'Controlling target brain regions by optimal selection of input nodes' has been provisionally accepted for publication in PLOS Computational Biology.

Best regards,

Hayriye Cagnan

Academic Editor

PLOS Computational Biology

Lyle Graham

Section Editor

PLOS Computational Biology

We thank the authors for the careful revision of their manuscript.

---

## [Editor Report · Acceptance letter]

20 Dec 2023

PCOMPBIOL-D-23-00955R1 

Controlling target brain regions by optimal selection of input nodes

Dear Dr Allegra,

I am pleased to inform you that your manuscript has been formally accepted for publication in PLOS Computational Biology. Your manuscript is now with our production department and you will be notified of the publication date in due course.

With kind regards,

Dorothy Lannert
